



# Rhizosphere to the atmosphere: contrasting methane pathways, fluxes and geochemical drivers across the terrestrial-aquatic wetland boundary

Luke C. Jeffrey[1,2], Damien T. Maher[1,2,3], Scott Johnston[1], Kylie Maguire[1], Andrew D.L. Steven[4] and Douglas R. Tait[1,2].

[1]SCU Geoscience, Southern Cross University, PO Box 157, Lismore, NSW 2480, Australia.

[2]National Marine Science Centre, Southern Cross University, PO Box 4321, Coffs Harbour, NSW 2450, Australia.

[3]School of Environment, Science and Engineering, Southern Cross University, Lismore, NSW 2480, Australia

[4]CSIRO Oceans and Atmosphere, Queensland Biosciences Precinct, University of Queensland, 306 Carmody Rd, St Lucia, Brisbane 4067, Australia

Corresponding author: luke.jeffrey@scu.edu.au

**Key Words:**

Diffusion

Ebullition

Sediment redox

Coastal acid sulphate soils

Sulphate reduction

Iron reduction

Carbon cycle





***Abstract***
*Although wetlands represent the largest natural source of atmospheric $CH_4$, large*
*uncertainties remain regarding the global $CH_4$ flux. Wetland hydrological oscillations*
*contribute to this uncertainty, dramatically altering wetland area, water table height, soil*
*redox potentials and $CH_4$ emissions. This study compares both terrestrial and aquatic $CH_4$*
*fluxes over two distinct seasons in both permanent and seasonal remediated freshwater*
*wetlands in subtropical Australia. We account for aquatic $CH_4$ diffusion and ebullition rates,*
*and plant-mediated $CH_4$ fluxes from three distinct vegetation communities, thereby examining*
*seasonal, diurnal and intra-habitat variability. $CH_4$ emission rates were related to underlying*
*sediment geochemistry. For example, distinct negative relationships between $Fe(III)$ and $SO_4^{2-}$*
*and $CH_4$ fluxes were observed, whereas distinct positive trends occurred between $CH_4$*
*emissions and $Fe(II)$ / AVS, where sediment $Fe(III)$ and $SO_4^{2-}$ were depleted. The highest $CH_4$*
*emissions of the seasonal wetland were measured during flooded conditions and always during*
*daylight hours, which is consistent with soil redox potential and temperature being important*
*co-drivers of $CH_4$ flux. The highest $CH_4$ fluxes were consistently emitted from the permanent*
*wetland (1.5 to 10.5 mmol $m^{-2}$ $d^{-1}$), followed by the Phragmites australis community within the*
*seasonal wetland (0.8 to 2.3 mmol $m^{-2}$ $d^{-1}$), whilst the lowest $CH_4$ fluxes came from a region of*
*forested Juncus sp. (-0.01 to 0.1 mmol $m^{-2}$ $d^{-1}$) which also corresponded with the highest*
*sedimentary $Fe(III)$ and $SO_4^{2-}$. We suggest that wetland remediation strategies should consider*
*geochemical profiles to help to mitigate excessive and unwanted methane emissions, especially*
*during early system recovery periods.*





## 1.0 Introduction

Wetlands are considered one of the most valuable ecosystems on Earth (Costanza et al., 2014). They are biodiversity hotspots that provide ecosystem services such as water filtration, sediment trapping, floodwater retention and carbon (C) storage (Bianchi, 2007). Wetlands account for ~5.5% of terrestrial surfaces (Melton et al., 2013) and have been estimated to store from ~4% (Bridgham et al., 2014) to ~30% (Mitsch et al., 2013) of Earth's estimated 2500 Pg soil C pool (Lal, 2008). Pristine wetlands have long been considered net C sinks due to their high rates of productivity and low rates of decomposition (Mitsch et al., 2013); however due to their waterlogged nature and anaerobic soils, wetlands are ideal environments for the production of methane ($CH_4$), a potent greenhouse gas. As such, wetlands are recognised as Earth's largest natural source of $CH_4$ to the atmosphere ($185 \pm 21$ Tg C yr$^{-1}$) (Saunois et al., 2016).

Resolving the drivers, pathways and effects of seasonal weather oscillations on wetland $CH_4$ sink or source behaviours is important to enable more accurate climate model projections and to reduce uncertainties in the global wetland $CH_4$ budget (Kirschke et al., 2013; Saunois et al., 2016). Mitsch et al. (2013) estimated that the average ratio of freshwater wetland $CO_2$ sequestration to $CH_4$ emissions was 19.5:1. As $CH_4$ is 34 times more potent than carbon dioxide ($CO_2$) over a 100 year time scale (Stocker et al., 2013), this suggests that many freshwater wetlands may have a net positive radiative forcing effect on climate (Hernes et al., 2018). However, variability in geomorphology, wetland maturity, salinity and underlying geochemical composition all contribute to variable $CH_4$ dynamics (Bastviken et al., 2011; Poffenbarger et al., 2011; Whiting & Chanton, 2001). The lack of spatially-resolved wetland $CH_4$ emission data, as well as the limited number of studies constraining the multiple wetland $CH_4$ flux pathways (i.e. ebullition, diffusion and plant-mediated) coupled with ongoing anthropogenic conversion of wetland systems (Bartlett & Harriss, 1993; Neubauer & Megonigal, 2015; Saunois et al., 2016) further contribute to the uncertainties around $CH_4$ regional to global scale budgets.

Extensive clearing and drainage of many coastal wetlands has occurred over the previous two centuries in order to accomodate agriculture, aquaculture and urban development (Armentano & Menges, 1986; Villa & Bernal, 2018; White et al., 1997). Drained wetlands can lead to rapid soil organic matter oxidation, and transform systems to net $CO_2$ sources (Deverel et al., 2016; Pereyra & Mitsch, 2018). Drainage systems can also reduce wetland inundation



periods and alter sediment redox-dependant geochemistry and microbially-mediated reactions
(Johnston et al., 2014), particularly those involving bioavailable iron (Fe(III)), sulphate ($SO_4^{2-}$
) and nitrate ($NO_3^-$). Importantly, anaerobic carbon metabolism employing these terminal
electron acceptors (Fe(III), $SO_4^{2-}$, $NO_3^-$) competes thermodynamically with methanogenic
bacteria and archaea and thereby can inhibit $CH_4$ production (á Norði & Thamdrup, 2014;
Burdige, 2012; Karimian et al., 2018; Lal, 2008). With increasing value now placed on the
ecosystems services provided by wetlands, many degraded systems are now undergoing
remediation and re-flooding (Johnston et al., 2014). However, the ecosystem benefits, such as
enhanced biodiversity and water quality, may come at a price in the form of high initial $CH_4$
flux rates, and predicted net radiative forcing for several centuries post-remediation - thus
posing a 'biogeochemical compromise' (Hemes et al., 2018; Lal, 2008).

Within Australia, it has been estimated that more than 50% of natural wetlands have

been lost to land use change, drainage and degradation since European settlement (Finlayson
& Rea, 1999; ANCA, 1995). This equates to an estimated ~1.2 Pg C emitted to the atmosphere
through oxidation of soil organic carbon (Page & Dalal, 2011). Much of eastern Australia's
freshwater coastal wetlands are underlain by Holocene derived sulphidic sediments (i.e pyrite
– $Fe_2S$, known as coastal acid sulphate soils; CASS) formed during periods of higher sea levels
(Walker, 1972; White et al., 1997). When CASS are drained, pyrite is oxidised, producing
sulphuric acid ($H_2SO_4$). This results in highly acidic soils with pH levels as low as 3 (Johnston
et al., 2014; Sammut et al., 1996). After rainfall events, groundwater transports $H_2SO_4$ from
the CASS landscapes into nearby creeks and estuaries (Sammut et al., 1996). The low pH
groundwater discharge also mobilises iron and aluminium, fuels aquatic deoxygenation, and
can lead to large fish kills and degradation of infrastructure (Jeffrey et al., 2016; Johnston et
al., 2003; White et al., 1997; Wong et al., 2010). Drained CASS wetlands typically contain
abundant reactive Fe(III) and exhibit complex sulphur and Fe cycling (Boman et al., 2008;
Burton et al., 2011; Burton et al., 2006). Wetland iron and sulfur cycling can profoundly
influence $CH_4$ production and consumption via a series of complex redox reactions coupled
with organic matter mineralisation (Holmkvist et al., 2011; Sivan et al., 2014). As such,
terminal electron acceptor availability is critical when considering wetland remediation and the
biogeochemical compromise paradigm.
Here we assess $CH_4$ emissions rates from a remediated freshwater CASS wetland in
subtropical eastern Australia, and compare fluxes from the permanent wetland and the adjacent
seasonal wetland ecotypes. We hypothesize that wetland $CH_4$ emissions will differ



significantly between the seasons and between the four wetland communities. We account for three atmospheric flux pathways for methane; ebullition, diffusion and plant-mediated fluxes, over diurnal cycles and within different seasons. CH$_4$ fluxes were also assessed in relation to the underlying soil properties, including sulphate, reactive iron III and iron II, acid volatile sulphur, chloride and organic carbon.

## 2.0 Methods

### 2.1 Study site

Cattai Wetland is located on the mid-coast of New South Wales, Australia. The reserve covers 500 hectares, featuring a shallow permanent wetland covering an area of approximately 16 hectares that is adjacent to a seasonal wetland and floodplain located to the south (Fig. 1). Both sites discharge into the nearby Coopernook Creek, a tributary of the larger Manning River estuary. The site was extensively cleared and low-lying areas drained during the early 1900's in order to aid agriculture and development in the region. As a result of this anthropogenic drainage, the oxidation of CASS produced sulphuric acid and episodic acidic discharge to adjacent creeks for many years (Tulau, 1999). To ameliorate acidic discharge, the natural hydrology of the site was restored in 2003 through the decommissioning of agricultural drains and removal of floodgates. Re-flooding of the CASS landscape has reduced the production of sulphuric acid, acid discharge and aluminium and iron mobilisation, hence improving the downstream water quality (GTCC, 2014).

The region receives a mean annual rainfall of 1180 mm with the majority falling during early autumn with an average maximal monthly rainfall in March (152 mm). The lowest rainfall generally occurs during the winter months with average minimal rainfall during September (60 mm). Average minimum and maximum summer temperatures range from 17.6 °C to 29 °C (January) and in winter range from 5.9 °C to 18.5 °C (July) (BOM, 2018). The dominant vegetation type within the permanent wetland is an introduced waterlily species (*Nymphaea capensis*), while the fringes of the wetland consist of wetland tree species; *Casuarina* sp. and *Melaleuca quinquenervia*. The seasonal wetland to the south is dominated by the sedge; *Juncus kraussii* (Veg A) and features scattered stands of *Phragmites australis* (Veg B) with areas of slightly higher elevation dominated by *Juncus kraussii* below *Casuarina sp.* (Veg C) (Fig. 1).





## 2.2 The aquatic CH₄ flux of the permanent wetland


To quantify $CH_4$ ebullition rates, up to 12 ebullition domes were deployed during two
distinct seasons (detailed below) at ~20 m intervals along a longitudinal transect, from the edge
of the permanent wetland towards the centre. Each dome was carefully suspended below the
water level by flotation rings, ensuring minimal disturbance of sediment and the water column.
Gas samples were extracted from the headspace of each dome using a 300 mL gas tight syringe
at periods of ~48 h. The volume was recorded and each sample then diluted using ambient air
(1:729 ratio) and analysed in situ using a using a manufacturer calibrated cavity ring-down
spectrometer (Picarro G2201-*i*) to determine $CH_4$ concentrations (ppm). Diffusive $CH_4$ fluxes
from the permanent wetland were measured using a floating chamber with a portable
greenhouse gas analyser (UGGA, Los Gatos Research). To account for spatial and temporal
variability, measurements were conducted during both day-time and night-time, and sampling
within vegetated areas featuring lilies (*Nymphaea capensis*), forested areas (*Melaleuca* sp.)
and in areas where no aquatic vegetation was present (i.e. open water). A total of 39 $CH_4$
floating chamber incubations averaging ~8 minutes in duration were recorded over the two
campaigns. The average $r^2$ value of linear regressions of $CH_4$ concentrations versus time during
chamber incubations was $0.97 \pm 0.05$. One chamber measurement was disregarded as an outlier
(as it was more than three times the standard deviation of the mean) and any chambers capturing
ebullition bubbles (determined by a nonlinear increase in concentration) were also disregarded.
The seasonal ebullition and diffusive $CH_4$ flux methods and measurements from the permanent
wetland have previously been reported elsewhere (Jeffrey et al. submitted).

## 2.3 Plant-mediated CH₄ fluxes


Simultaneous time series chamber experiments were conducted over ~24 hours to
measure $CH_4$ fluxes during each season from the three different wetland vegetation ecotypes.
These ecotypes were *Juncus kraussii* (Veg A), *Phragmites australis* (Veg B) and *Juncus*
*kraussii* amongst *Casuarina sp.* forest (Veg C) (Fig. 1). In each ecotype, 65 x 65 x 30 cm
acrylic bases were installed four months before the first time series experiment, to minimise
disturbance to the sediment profile and vegetative rhizosphere. Vegetative flux chambers were
constructed of an aluminium frame with clear Perspex walls and roof that matched the areal
footprint of the pre-inserted acrylic bases. The chambers were 100 cm, 150 cm and 50 cm high
for at Veg A, B and C respectively. The custom sizes were tailored for the different vegetation



heights, whilst minimising chamber volume as much as possible. Each chamber was leak-tested
under laboratory conditions prior to fieldwork.
Before each field incubation, chambers were flushed with atmospheric air then
carefully lowered over the vegetation and onto the acrylic base ensuring an air tight seal. A
small fan circulated internal air within each chamber. Air within the chamber was pumped
through a closed loop from the top of the chamber using gas tubing (Bevaline), passing through
a drying agent (Drierite desiccant) and then analysed in situ using a calibrated cavity ring-down
spectrometers (Picarro G2201-*i* or LosGatos), recording the flux rate of $CH_4$ (ppm/sec). The
gas flow was returned near the base inside each vegetation chamber closing the loop.
Vegetation incubation times ranged from 6 to 15 minutes depending on the flux rate and were
taken from triplicate sites to account for heterogeneity within each ecotype. The daytime
measurements (after sunrise) were measured at ~10 minute intervals whilst night time
measurements (after sunset) were taken at ~4 hourly intervals. $CH_4$ fluxes from the adjacent
exposed sediments or shallow overlying water at each site were also measured at ~4 hourly
intervals to determine the influence and role of plant-mediated $CH_4$ fluxes compared to non-
vegetated $CH_4$ fluxes. Light and temperature loggers (Onset Hobo) measured the changes in
diurnal air temperature (°C) and photosynthetically active radiation (PAR) at each site.

**2.4 Soil geochemistry and redox conditions**
A water logger (Minidiver) was deployed in the permanent wetland before the first
campaign to monitor changes in water depth (cm) and temperature (°C). Field pH ($pH_F$) and
the redox potential ($Eh_F$; reported against standard hydrogen electrode) were determined in
situ, by directly inserting the electrode into the soils (5 cm depth, 8 replicates) at each site. A
composite sampling approach (3 cores) was used to collect sediment samples from each site,
to determine organic C content, $Fe(III)_{HCl}$, $Fe(II)_{HCl}$, Cl, $SO_4^{2-}$ and acid volatile sulphur (AVS).
The cores were extracted in December 2016, by inserting a 4.0 cm diameter acrylic tube into
the sediment to a depth of up to 50 cm. Cores were immediately sectioned into 2 cm increments
to a depth of 20 cm, and 5 cm increments thereafter, ensuring higher vertical resolution in the
organic rich near-surface sediments. Samples were immediately placed into air-tight bags, then
frozen within 12 hr of collection at -16°C in a portable freezer and transferred to -80°C freezer
in the laboratory. Frozen samples were thawed in an oxygen-free anaerobic chamber (1-5% $H_2$





in N$_2$), using an oxygen consuming palladium (Pd) catalyst. The defrosted samples were
homogenised using a plastic spatula.

AVS content was determined by adding 1-2 g of wet sediment with 6 M HCl:1 M L-

ascorbic acid. The liberated H$_2$S was captured in 5 ml of 3% Zn acetate in 2 M NaOH and then
quantified using iodometric titration. The reactive Fe fractions were determined using a
sequential extraction procedure optimised for acid sulphate soils based on Claff et al. (2010).
Poorly crystalline solid-phase Fe (II) and Fe (III) were determined by extracting 2 g wet sub-
samples with cold N$_2$-purged 1 M HCl for four hours. Aliquots of 0.45 µm-filtered extract were
analysed for Fe (II) [Fe(II)$_{HCl}$] and total Fe [Fe$_{HCl}$] using the 1,10-phenanthroline method with
the addition of hydroxylammonium chloride for total Fe (APHA, 2005). The Fe(III) [Fe(III)$_{HCl}$]
was determined by the difference of [Fe$_{HCl}$] – [Fe(II)$_{HCl}$]. Total organic carbon (TOC) and total
S (S$_{Tot}$) were determined via a LECO CNS-2000 carbon and sulfur analyser. Chloride and
sulfate concentrations were measured using filtered (0.45 µm) aliquot from a 1:5 water extract
of freshly defrosted wet soil, as per Rayment and Higginson (1992) via ion chromatography
using a Metrosep A Supp4-250 column, an RP2 guard column and eluent containing 2 mM
NaHCO$_3$, 2.4 mM Na$_2$CO$_3$ and 5% acetone, in conjunction with a Metrohm MSM module for
background suppression.

**2.5 Calculations**

Both the air-water and vegetative CH$_4$ fluxes were calculated for the chamber

deployments in the permanent wetland and seasonal wetland using the equation:

$F = (s(V/RT_{air}A))t$                  (1)

where $s$ is the regression slope for each chamber incubation deployments (ppm sec$^{-1}$), $V$ is the
chamber volume (m$^3$), $R$ is the universal gas constant, $T_{air}$ is the air temperature inside the
chamber ($K$), $A$ is the surface area of the chamber (m$^2$) and $t$ is the conversion factor from
seconds to day, and to mmol.
Ebullition rates (E$_b$) (mmol m$^{-2}$ d$^{-1}$) were calculated using the equation:

$E_b = ([CH_4].CH_{4Vol.})/ A.V_m.Td$              (2)





where [$CH_4$] is the $CH_4$ concentration in the collected gas (%), $CH_{4Vol.}$ is the gas volume
sampled (L), A is the funnel area (m$^2$), $V_m$ is the molar volume of $CH_4$ at in situ temperature
(L) and Td is deployment time (days).

**3.0 Results**
**3.1 Hydrological Conditions**

Prior to the first campaign in April 2017 (C1), an extreme hot/drying summer period

occurred during early 2017 (Fig. 2). This resulted in an average wetland water column
temperature of $23.3 \pm 0.7$ °C and a water depth in the permanent wetland as low as ~7.3 cm,
with exposed sediments along the wetland perimeter during the preceding month. Total rainfall
for the two weeks prior to C1 was 342 mm, with an additional 35 mm of rain occurring during
C1 fieldwork (Fig. 2) thus raising the water column depth in the permanent wetland to 77.2 cm
in less than four weeks. This C1 deployment was therefore categorized as the 'post-
dry/flooded' period, where air temperatures ranged from 13.3 to 22.8 °C and the average water
column temperature in the permanent wetland was $20.4 \pm 0.5$ °C. The second fieldwork
campaign was conducted in September 2017 (C2) under cool/drying conditions, where air
temperatures ranged from as low as 3.4 °C to 34.9 °C (Fig. 2), with cooler average water
temperatures $12.6 \pm 0.4$ °C in the permanent wetland (Fig. 2). The depth of the permanent
wetland at this time had dropped slightly to ~33 cm (Fig. 2).

**3.2 Permanent and Seasonal Wetland $CH_4$ fluxes**

The vegetation time series revealed diurnal variability of plant-mediated $CH_4$ emissions

occurred at most ecotypes, with the highest $CH_4$ fluxes occurring during daytime around mid-
day and the lowest $CH_4$ fluxes during the night time (Fig. 3, Table 1). The lowest $CH_4$ fluxes
were found at Veg C with a net negative $CH_4$ flux observed during C2 time series. The $CH_4$
sediment fluxes measured amongst each vegetation time series were consistently much lower
than the plant-mediated $CH_4$ fluxes indicating that the vegetation was indeed the main conduit
for $CH_4$ to the atmosphere (Fig. 3, Table 1). The $CH_4$ fluxes were highly variable between the
replicates at each site. Temperature and PAR followed similar diurnal trends to each other and
had positive correlations to $CH_4$ emissions (Fig. 3).





$CH_4$ fluxes from the three vegetation types were higher in C1 than C2 (Fig. 4, Table 1).
The highest $CH_4$ fluxes in each of the vegetation types always occurred during the daytime
(Fig. 4, Table 1). *Phragmites sp.* (Veg B) consistently emitted the highest $CH_4$ fluxes ($2.27 \pm$
$1.42$ mmol m$^{-2}$ d$^{-2}$ during C1 and $0.77 \pm 0.46$ mmol m$^{-2}$ d$^{-1}$ during C2). The Veg C ecotype
within the seasonal wetland consistently produced the lowest $CH_4$ fluxes of all sites, with a net
negative flux occurring during C2 ($-0.01 \pm 0.08$ mmol m$^{-2}$ d$^{-1}$).
The permanent wetland showed an inverse trend with seven-fold higher diffusive fluxes
during the cool/drying C2 ($10.46 \pm 15.81$ mmol m$^{-2}$ d$^{-1}$) compared to the post-dry/flooded C1
($1.49 \pm 2.75$ mmol m$^{-2}$ d$^{-1}$), while the ebullition rates were similar during both seasons (Fig. 4,
Table 1). Overall, the plant mediated $CH_4$ fluxes from the three seasonal wetland vegetation
ecotypes (Veg A, B and C) were within the range of aquatic fluxes measured from the
permanent wetland for the post-dry/flooded C1 time series, but not for the cool/drying C2 time
series, when the permanent wetland $CH_4$ fluxes were much higher (Fig. 4).

### 3.3 Sediment core profiles and soil redox potentials

Average concentrations from soil cores (Table 1, Fig. 5) were based upon the top 20
cm of the profile, where the highest organic carbon concentrations were found. This upper
rhizosphere depth zone is assumed to be an active area of carbon metabolism and $CH_4$
production and consumption (Nedwell & Watson, 1995). The Fe(III)$_{HCl}$ concentrations were
greater than Fe(II)$_{HCl}$ at all three seasonal wetland sites, however the permanent wetland
showed an opposite trend with low concentrations of both Fe(III) ($5.6 \pm 10.7$ mmol kg$^{-1}$) and
SO$_4^{2-}$ ($1.5 \pm 1.0$ mmol kg$^{-1}$) (Fig. 5, Table 1). The highest average concentrations of Fe(III)$_{HCl}$
were found at the Veg C site ($204.0$ mmol kg$^{-1}$) and highest and similar concentrations of SO$_4^{2-}$
were in Veg B and Veg C sediments ($45.4 \pm 41.0$ mmol kg$^{-1}$ and $43.3 \pm 16.7$ mmol kg$^{-1}$) (Fig.
5, Table 1). Net positive redox potential was found at all four sites during C1 (under post-dry/
flooded conditions) indicating a lag time between recent flooding and the onset of reducing
conditions. In contrast, a negative redox potential was found within the permanent wetland and
Veg B during C2, indicating reduced conditions under cool drying conditions (Table 1). The
TOC concentrations (%) were highest in the upper profiles and similar across all sites (Fig. 5,
Table 1) averaging $13.4 \pm 7.6\%$.



### 3.4 Temperature and PAR


Correlation plots for both temperature (ºC) and sunlight (PAR) versus $CH_4$ emissions
from the three vegetation ecotypes showed no distinct relationships with the exception of Veg
B during C2 for PAR ($r^2$=0.18, p<0.01) and temperature (ºC) ($r^2$=0.35, p<0.001). No clearer
trends were observed by combining all site measurements, nor separating daytime fluxes and
drivers from night time fluxes and drivers.

### 4.0 Discussion


### 4.1 Geochemistry of the CASS landscape


Sediment profiles provide insights to the historical geochemical changes that have
occurred across the CASS landscapes of the four Cattai Wetland sites (Fig. 5). If we assume
that relatively uniform deposition of late Holocene materials occurred, the differences between
present day profiles are related to historical changes in hydrology and land use, topographic
elevation, geochemical trajectories and vegetative carbon inputs. For example, the permanent
wetland shows distinct differences to the adjacent seasonal wetland sites, with divergent
geochemical signatures of both iron and sulphate that reflect the sustained inundation (Table
1, Fig 5). The permanent wetland had significantly lower Fe(III) (p<0.001) and 11 to 30 fold
lower $SO_4^{2-}$ concentrations within the upper soil profile compared to the seasonal wetland. The
ratio of Fe(III)$_{HCl}$ to Fe(II)$_{HCl}$ from the flooded soils of the permanent wetland was 0.03,
indicating the sediments were almost completely depleted of Fe(III). Under reducing
conditions where there is low $SO_4^{2-}$ and little to no Fe(III) to competitively exclude
methanogenesis, $CH_4$ production becomes more favourable. Indeed, $CH_4$ production was on
average highest from the permanent wetland, especially when considering the duel $CH_4$
pathways of ebullition and air-water diffusion (Table 1).
In addition to sulphate reduction, some depletion of the sulphur pool from the
permanent wetland may have occurred due to drainage exports of sulphuric acid ($H_2SO_4$)
discharging from the CASS landscape throughout the last century. Alternatively, reducing
conditions induced by re-flooding freshwater wetlands is known to encourage the re-formation
of AVS and pyrite ($FeS_2$) and produce alkalinity, thereby attenuating acid production and
discharge (Burton et al., 2007; Johnston et al., 2014; Johnston et al., 2012) and reducing the
total $SO_4^{2-}$ pool of CASS landscapes. While the AVS concentrations found within the





permanent wetland (up to 18.5 µmol g$^{-1}$) were a result of sulphate reduction induced by CASS wetland restoration, they nonetheless represent a relatively volatile form of sulphur, which is at risk of rapid oxidation during drought periods (Johnston et al., 2014; Karimian et al., 2017). The AVS concentrations of the permanent wetland sites were more than 20-fold higher than the three adjacent seasonal wetland sites, and represent a potentially volatile by-product and consequence of re-flooding CASS soil landscapes, in addition to leading to increases of CH$_4$ emissions (Table 1).

The soil profile from the seasonal wetland Veg C habitat featured abundant Fe(III)$_{HCl}$ (Fe(III)$_{HCl}$ to Fe(II)$_{HCl}$ ratio of 136) and also SO$_4^{2-}$. This was associated with the lowest fluxes of CH$_4$ for both seasonal sampling periods (Fig. 5, Table 1). Relatively low CH$_4$ fluxes from Veg C are likely due to the more oxidising conditions present at this site and the surfeit of thermodynamically favourable terminal electron acceptors (i.e. Fe(III) and SO$_4^{2-}$), which would competitively exclude organic matter degradation by methanogenic archaea (Postma & Jakobsen, 1996).

At the other seasonal wetland sites (Veg A and B), the average Fe(III) and SO$_4^{2-}$ concentrations were intermediate, (i.e. lower than Veg C, but higher than the permanent wetland), although in the upper profile Veg B had more SO$_4^{2-}$ while Veg A had more Fe(III) (Fig. 5, Table 1). CH$_4$ flux values from these sites were also intermediate (Table 1). Sediment profiles from both Veg A and Veg B indicated a degree of Fe reduction based on the ratio of Fe(III):Fe(II) which were 7.2 and 3.6 respectively. The redox potentials from Veg B during both C1 and C2 seasons (9.6 mV and -89.0 mV respectively) were consistently lower than Veg A during C1 and C2 seasons (46.5 mV and 12.0 mV respectively), which is consistent with the more reducing conditions encouraging CH$_4$ production in Veg B habitat. Further, as iron reduction yields more free energy than SO$_4^{2-}$ reduction (which yields more free energy than methanogenesis) (Burdige, 2012), then Fe reduction at Veg A may outcompete CH$_4$ production ahead of SO$_4^{2-}$ reduction at Veg B, which may help explain some of the differences in CH$_4$ production between the two sites.

Regression analysis and Spearman rho coefficients summarise the spatial trends occurring between the average sediment parameters versus seasonal CH$_4$ fluxes from the different sites (Fig. 7). Positive significant trends occurred for Fe(II), AVS and the Cl:SO$_4^{2-}$ ratios with CH$_4$ flux rates ($r_s$=0.88, p<0.01) supporting our hypothesis that reducing conditions and a smaller pool of sediment Fe(III) and SO$_4^{2-}$ facilitate higher CH$_4$ production rates.





Alternatively, the negative trends observed between soil redox potentials, $SO_4^{2-}$, Fe(III) and
$CH_4$ fluxes affirm that the abundance of thermodynamically favourable terminal electron
acceptors plays a role in attenuating $CH_4$ production at each site.

**4.2 Plant-mediated $CH_4$ fluxes from the seasonal wetland**

Plant-mediated $CH_4$ fluxes were highest during C1 under post-dry/flooded conditions

with 20-30 cm of standing waters in the seasonal wetland (Table 1). While waterlogged
conditions are an obvious driver of higher $CH_4$ production rates from saturated sediments in
addition to the geochemical differences (previously discussed), other drivers which may
explain these trends include differences in diurnal variability in temperature, PAR and plant
physiology, which may influence $CH_4$ gas transport pathways.

In vegetated seasonal wetlands, plant-mediated gas transport is recognised as a

dominant pathway for $CH_4$ emission to the atmosphere and accounts for up to 90% of total
wetland fluxes (Sorrell & Boon, 1994; Whiting & Chanton, 1992). For plant survival in near-
permanent inundation environments, oxygen transport occurs via the araenchyma downwards
to the rhizome. This increases the plant performance by mitigating (i.e. oxidising) the
accumulation of phytotoxins such as sulphides and reducing metal ions around the roots
(Armstrong & Armstrong, 1990; Armstrong et al., 2006; Penhale & Wetzel, 1983). As oxygen
transfer to the rhizosphere occurs, an exchange of sedimentary $CH_4$ can be efficiently
transported from the rhizosphere to atmosphere, bypassing sedimentary oxidative processes
along the way (Fig. 8). This process in plants can be either convective (i.e. pressurised) or via
passive diffusive gas flow, both of which are adaptive traits of many wetland species
(Armstrong & Armstrong, 1991; Konnerup et al., 2011).

During both seasons the highest $CH_4$ fluxes from seasonal wetland vegetation were

emitted from *Phragmites australis* (Veg B) and always occurred during daylight (Table 1, Fig.
8). In *Phragmites australis* (Veg B), the presence of pressurised lacunar leaf culms drive a
mass flow of oxygen to the rhizome and back to the atmosphere via older (non-pressurised)
efflux culms (Henneberg et al., 2012; Sorrell & Boon, 1994). This process has been widely
studied in wetlands featuring this species, as it is one of the most productive and wide spread
flowering wetland species (Brix et al., 2001; Chanton et al., 2002; Clevering & Lissner, 1999;
Tucker, 1990). Kim et al. (1998) showed $CH_4$ emissions from *Phragmites australis* peaked
around midday and that daytime emissions were about 3-fold higher than night time emissions,





positively correlating with temperature and PAR. These were similar to our findings with
highest $CH_4$ fluxes of each seasonal time series occurring near midday (10:50 am during C1;
4.88 mmol $m^{-2}$ $d^{-2}$ and 12:15 pm during C2; 2.06 mmol $m^{-2}$ $d^{-2}$) (Fig. 3). We also found a
positive significant relationship between $CH_4$ flux and both temperature and PAR during C2
($r^2$=0.35, p<0.001 and $r^2$=0.18, p<0.01 respectively) (Fig. 6). The often high diurnal variability
in $CH_4$ fluxes from *Phragmites australis* occurs as convective gas transport increases
rhizospheric oxygen and $CH_4$ exchange via living culms during the daytime, whereas molecular
diffusion during the night time facilitates a more passive and lower $CH_4$ flux pathway through
dead culms (Armstrong & Armstrong, 1991; Chanton et al., 2002).

One possible reason $CH_4$ fluxes were lower from Veg A than Veg B despite their close

geographical location, may be due to the passive gas diffusion mechanism utilised by *Juncus*
*sp.* (Henneberg et al., 2012). Unlike the pressurised conductive gas flow mechanisms of Veg
B, many wetland rush species (such as Veg A) employ passive diffusive gas flow to survive
within water logging environments (Brix et al., 1992; Konnerup et al., 2011). Despite diffusion
being a less efficient gas transport mechanism (Konnerup et al., 2011), plant-mediated $CH_4$
diffusion is recognised as the dominant pathway for $CH_4$ emissions from many seasonal
wetland species. During C1 and C2, day time fluxes (diffusive) from Veg A were only 19%
and 33% higher than night time fluxes (diffusive). In comparison, at Veg B these day:night
ratios were almost triple this (67% and 94% higher) during the same periods. This may
potentially be due to the more efficient daytime conductive gas transfer pathway of $CH_4$
through Veg B (*Phragmites australis*) compared to the more passive diffusive $CH_4$ gas transfer
pathway of Veg A (*Juncus kraussii*). This suggests that non-pressurized pathways may result
in lower net rhizosphere-atmosphere gas exchange of $CH_4$ from seasonal wetland vegetation.

The *Juncus kraussii* below *Casuarina sp.* trees (Veg C) emitted nominal fluxes of $CH_4$

during both time series campaigns and was a net sink for $CH_4$ during C2 (Table 1, Fig. 8).
Although wetland trees have recently been shown to contribute significantly to $CH_4$ fluxes
from flooded environments (Pangala et al., 2017), we could not quantify or constrain the role
of trees as a conduit of methane to the atmosphere at this site. Regardless, there were clearly
lower $CH_4$ fluxes through the Veg C (*Juncus kraussii*) compared to the Veg A (*Juncus*
*kraussii*). As the species at ground level were identical, these differences are not related to
vegetative gas transport mechanisms, nor organic carbon content (Table 1). Shading by the
overhanging trees may inhibit the daytime diffusive $CH_4$ gas transport through Veg C
assumable to lower rates of photosynthesis, however PAR was only lower during C2 (Fig. 7)



and so does not appear to explain the CH$_4$ flux differences observed during C1. The differences
are therefore likely explained by the higher positive redox potentials (Table 1) and more
abundant thermodynamically favourable terminal electron acceptors (i.e. Fe(III) and SO$_4^{2-}$)
(Fig. 5) all of which can inhibit methane production within the sediments (Burdige, 2012).

**4.3 Permanent Wetland CH$_4$ fluxes**


Diffusive CH$_4$ fluxes from the permanent wetland varied considerably between
seasons; however, ebullition fluxes were similar (Table 1, Fig. 8). The highest seasonal CH$_4$
fluxes for both ebullition and diffusion (2.1 mmol m$^{-2}$ d$^{-1}$ and 10.5 mmol m$^{-2}$ d$^{-1}$ respectively)
occurred during C2 despite cooler conditions (Fig. 2. Fig. 8). This however was the opposite
trend to the seasonal wetland CH$_4$ fluxes (Table 1, Fig. 8). One reason may be due to the
antecedent hydrological conditions before C1 (Fig. 2). Jeffrey et al (submitted) reported that a
water level drawdown of the permanent wetland after a hot and drying summer period exposed
some of the permanent wetland sediments to oxidative conditions. This may have oxidised a
portion of the labile sedimentary carbon pool prior to C1 sampling of the permanent wetland,
therefore reducing the total CH$_4$ pool observed during C1 sampling. A lag time (ranging from
weeks to months) for recovery of the CH$_4$ pool post-drought has been observed in other systems
(Boon et al., 1997) and also during lab-based experiments (Freeman et al., 1992; Knorr et al.,
2008). This may explain the higher CH$_4$ fluxes during C2 when the system had had sufficient
time to recover, despite lower water column temperatures that would normally reduce
microbial metabolism rates. This hypothesis is also supported by the shift of net positive redox
potential of the permanent wetland during C1 (71.7 $\pm$ 65 mV), to a strong negative redox
potential during C2 (-216 $\pm$ 42 mV) indicating that there was a time lag for reducing conditions
to recover within the permanent wetland for C2. This highlights the critical role of antecedent
hydrological conditions and how dynamic weather oscillations of drought and floods (a
common occurrence of many Australian wetland systems), strongly influence the redox
potentials, soil geochemistry and ultimately CH$_4$ fluxes.

**4.4 Implications and conclusions**


Permanent wetland emissions account for the majority of the global wetland CH$_4$
budget however both subtropical systems and southern hemisphere systems are poorly





represented (Bartlett & Harriss, 1993; Bastviken et al., 2011) (Fig. 9). Further, the fluxes from
seasonal wetlands are poorly constrained (Pfeifer-Meister et al., 2018) due to their intermittent
nature and variability of intra-seasonal areal extent, which may compound why natural
wetlands have the largest uncertainty of the global methane budget (Kirschke et al., 2013;
Saunois et al., 2016). Although the temporal resolution of our study cannot be up scaled to
realistic annual estimates, our high resolution sampling strategy provided insights to daily $CH_4$
flux rates revealing distinct differences between different vegetation types across the terrestrial
aquatic wetland boundary. Our seasonal emissions rates were at the low end of the scale of
measurements made in southern hemisphere subtropical systems but within range of northern
hemisphere subtropical systems of similar latitudes (Fig. 9).

Although remediating degraded wetlands through re-flooding is a common technique

to improve biodiversity, increase C sequestration and improve downstream water quality issues
(Johnston et al., 2014; Johnston et al., 2004), our results propose a nuanced dilemma for land
use managers, as wetland restoration can have net positive radiative forcing effects on the
Earth's climate due to high rates of $CH_4$ production (Mitsch et al., 2013). This has also been
shown to be particularly high during early remediation periods (Hemes et al., 2018). Our results
suggest that seasonal wetlands emit less $CH_4$ on an areal basis than permanent wetlands, yet
carbon accumulation in these soils may be lower (Brown et al. (in publication)). Longer-term
studies over annual cycles encompassing seasonal drivers and $CH_4$ fluxes would further test
this hypothesis of the different drivers between seasonal and permanent wetland systems.

Our results also suggest that selective hydrological restoration of wetlands featuring

sediments with abundant thermodynamically favourable terminal electron acceptors (i.e.
Fe(III) or $SO_4^{2-}$) may be a (partial) biogeochemical solution (also suggested by Hemes et al.
(2018)) to both remediate degraded sites whilst simultaneously mitigating some $CH_4$
emissions. When Fe(III) and $SO_4^{2-}$ are abundant in anaerobic environments they provide
preferential terminal electron acceptors for microbial metabolism and thus limit
methanogenesis via competitive exclusion (Achtnich et al., 1995). However, high rates of
sulphate reduction coupled with Fe reduction can also lead to the accumulation of metal
sulphide minerals e.g. pyrite and AVS (Johnston et al., 2014). Under permanently saturated
and low oxygen conditions, metal sulphides will steadily accumulate and remain relatively
benign. However, if the saturated state of remediated sites cannot be maintained, AVS may
react with oxygen resulting in undesirable production of acidity and low pH conditions.
Therefore the remediation of wetlands for carbon storage should involve careful site selection





to both limit $CH_4$ production and to avoid redox related geochemical by-products with
detrimental environmental effects.
This study has highlighted how sediment geochemistry is intimately related to $CH_4$
production and consumption. While high sulphate and Fe(III) favour lower $CH_4$ production,
sites featuring more reducing conditions and depleted sulphate and Fe(III) favour the highest
$CH_4$ fluxes. Results reveal distinct differences between the areal $CH_4$ fluxes of four different
eco-types located within a remediated subtropical Australian wetland and indicate high
seasonal variability. By combining novel and well established techniques we delineated several
$CH_4$ pathways of both seasonal and permanent wetland sources (ebullition, diffusion and plant-
mediated pathways) and linked these to seasonal drivers. This provided evidence that soil
geochemistry is an important factor to consider for wetland remediation in the context of $CH_4$
production and mitigation strategies. The $CH_4$ emissions results were comparable to other
wetlands of similar latitudes and contribute important data for both the understudied southern
hemisphere wetlands and seasonal subtropical wetland ecotypes.

**Acknowledgements**

We would like to thank to Roz Hagan, Bob McDonnell, Zach Ford for assistance in the
field. We also thank Roz Hagan for processing the sediment cores, Isaac Santos and Ceylena
Holloway for technical support and Mid Coast Council for assistance. LCJ acknowledges
postgraduate support from CSIRO. This work was supported by funding from the Australian
Research Council. Graphic components used in conceptual model courtesy of the Integration
and Application Network, University of Maryland Centre for Environmental Science
(www.ian.umces.edu/symbols).





**Figures**

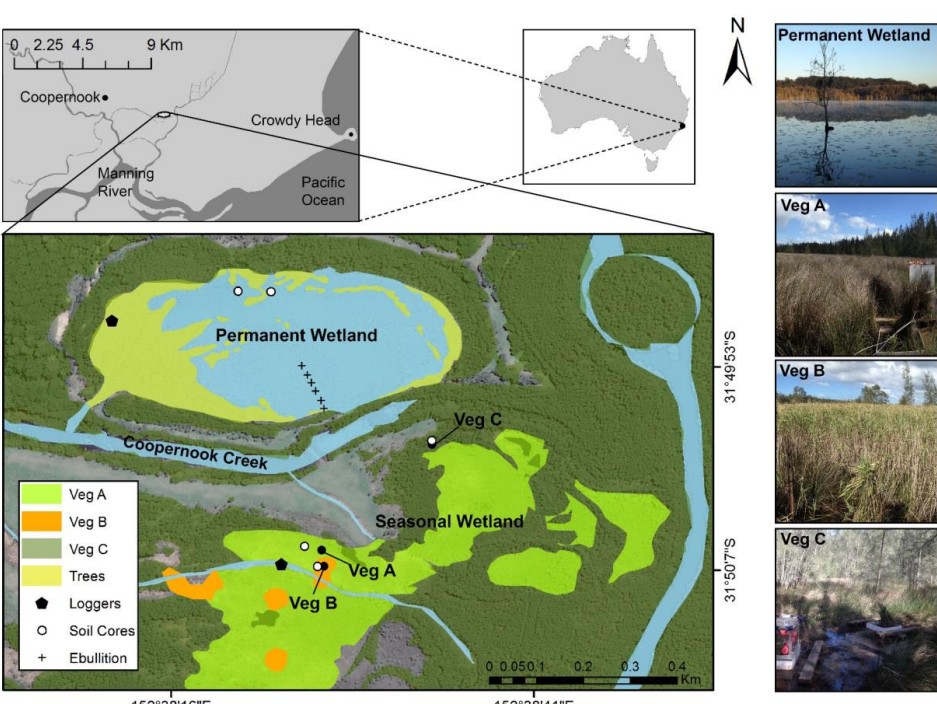


**Fig. 1** The seasonal wetland study sites consisting of Veg A (*Juncus kraussii*), Veg B
(*Phragmites australis*), Veg C (*Juncus kraussii* below *Casuarina sp.*) and the permanent
wetland indicating sediment coring sites, ebullition replicate transect, 24 h vegetation time
series sites and imagery of vegetation ecotypes.







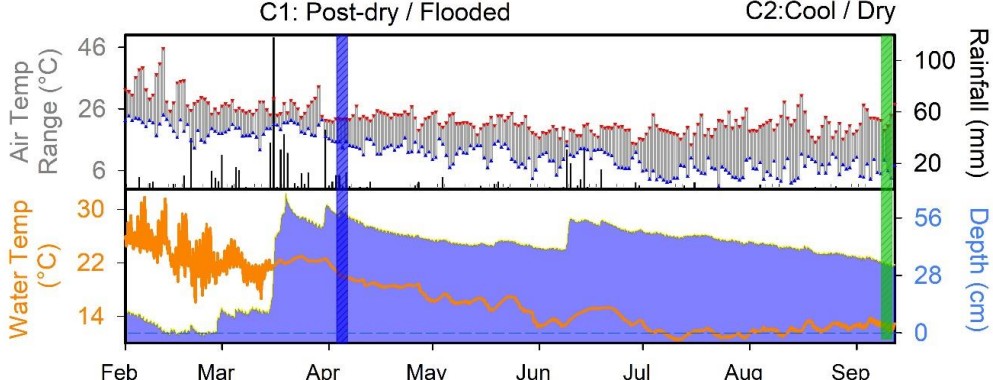


**Figure 2.** Hydrograph for the seven months of 2017 indicating daily rainfall, maximum/ minimum air temperature, water temperature and antecedent hydrology. Vertical coloured bands represent the two fieldwork campaigns.






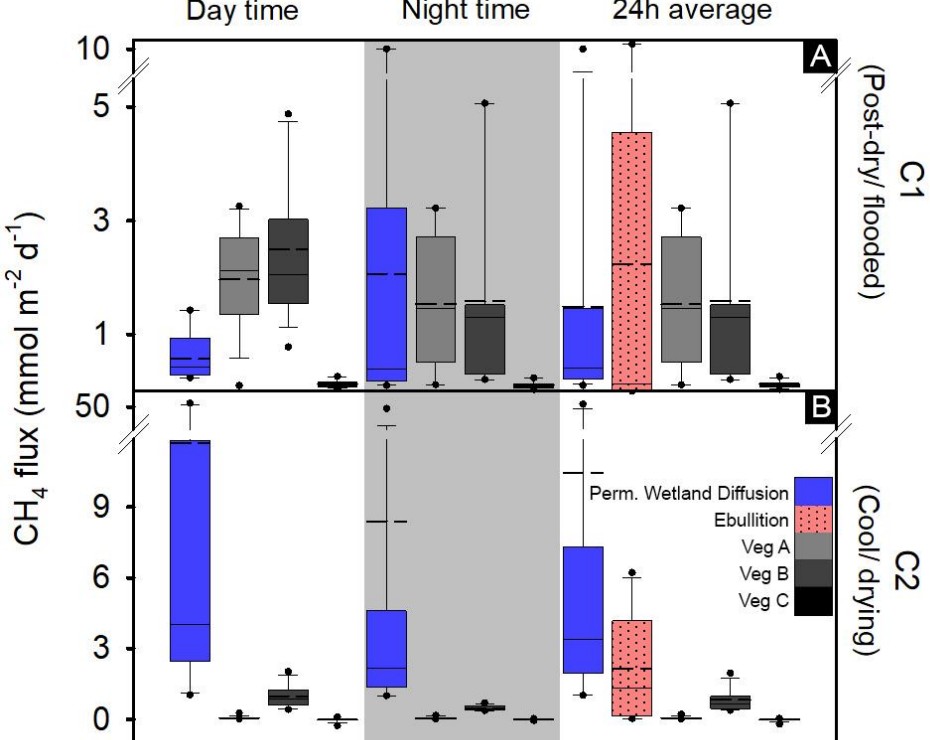


**Figure 3.** Simultaneous 24 h time series of vegetative $CH_4$ fluxes from the seasonal wetland
ecotypes at Cattai Wetland during C1: post-dry/flooded (Apr 2017) and C2: cool/drying
conditions (Sep 2017). The vertical error bars of the plant-mediated $CH_4$ flux (mmol m$^{-2}$ d$^{-1}$)
represent standard deviation of the triplicate time series measurements taken from each site and
horizontal bars represent the total aggregated time period represented by replicate chambers.
The grey shading indicates night-time. Note: Different y-axis scales for $CH_4$ to highlight
diurnal trends.



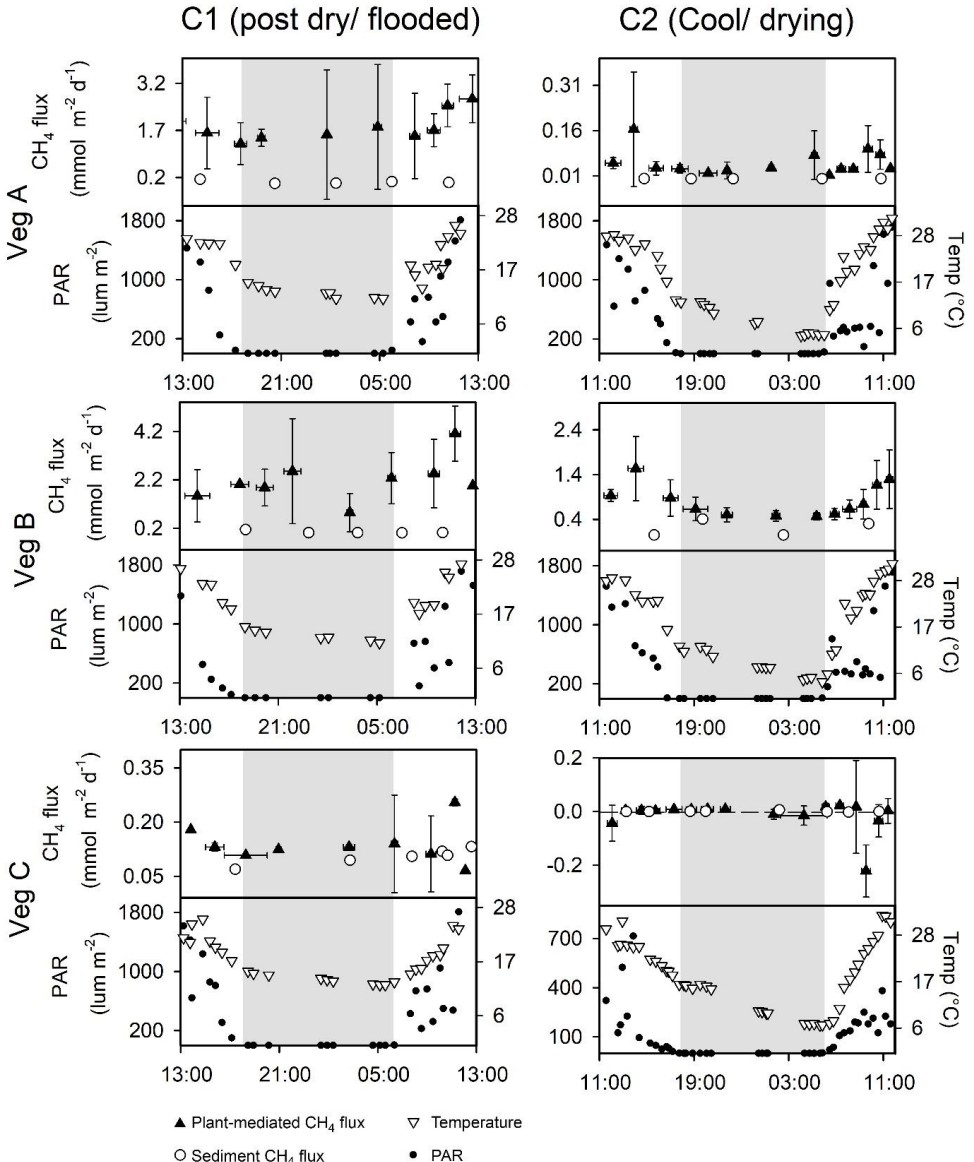

**Figure 4.** Seasonal fluxes of $CH_4$ from diurnal sampling and ebullition from the permanent

wetland and adjacent 24 h time series of the seasonal wetland vegetation types A, B and C.

Note: Dashed line represents the average, solid line represents the median and dots represent

$5^{th}$ and $95^{th}$ percentiles.





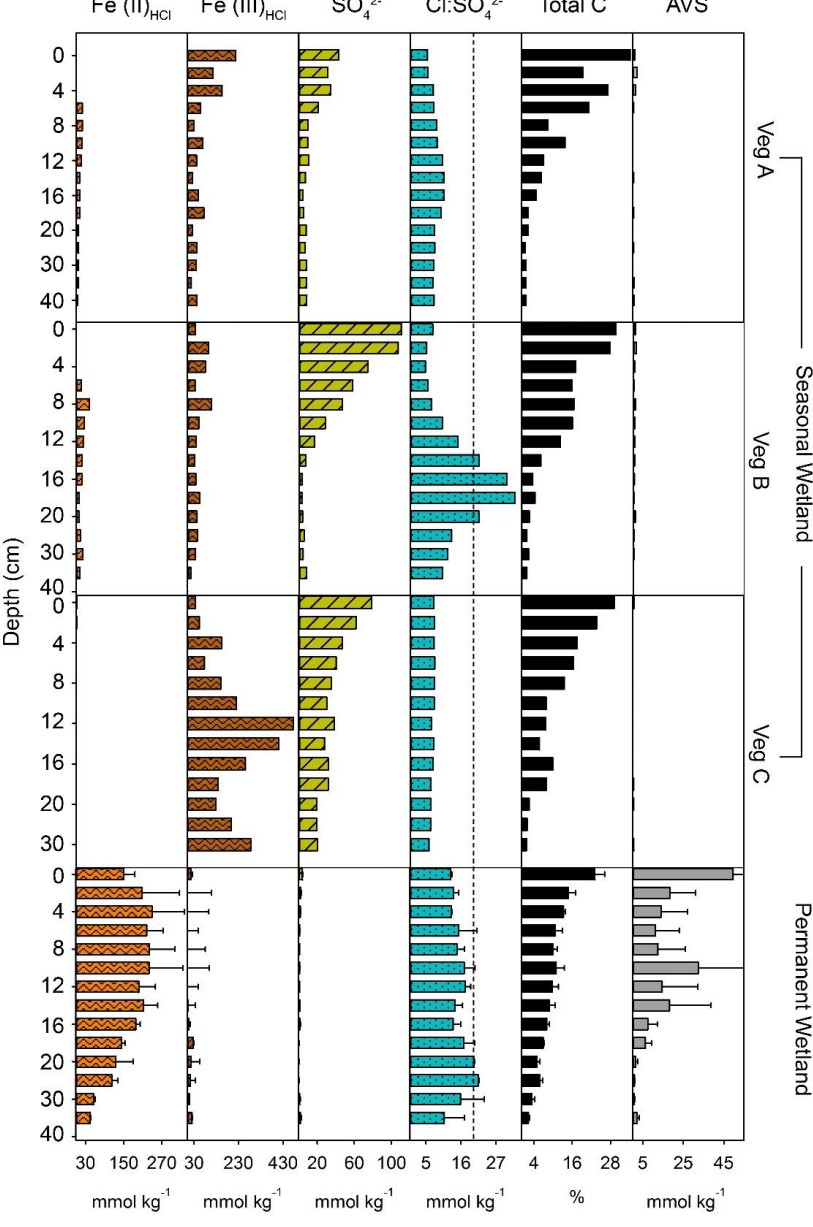

**Figure 5.** Soil profiles of the permanent and seasonal wetland sites indicating Fe(II)$_{HCl}$,
Fe(III)$_{HCl}$, SO$_4^{2-}$, Cl:SO$_4^{2-}$ (a proxy for depletion of marine-derived sulphate, where >20 is
broadly indicative of SO$_4^{2-}$ reduction and <8 CASS pyrite oxidation (Mulvey, 1993)), total C
and acid volatile sulphur (AVS). Note: The permanent wetland profiles are averages from two
adjacent sites with error bars representing the standard deviation.




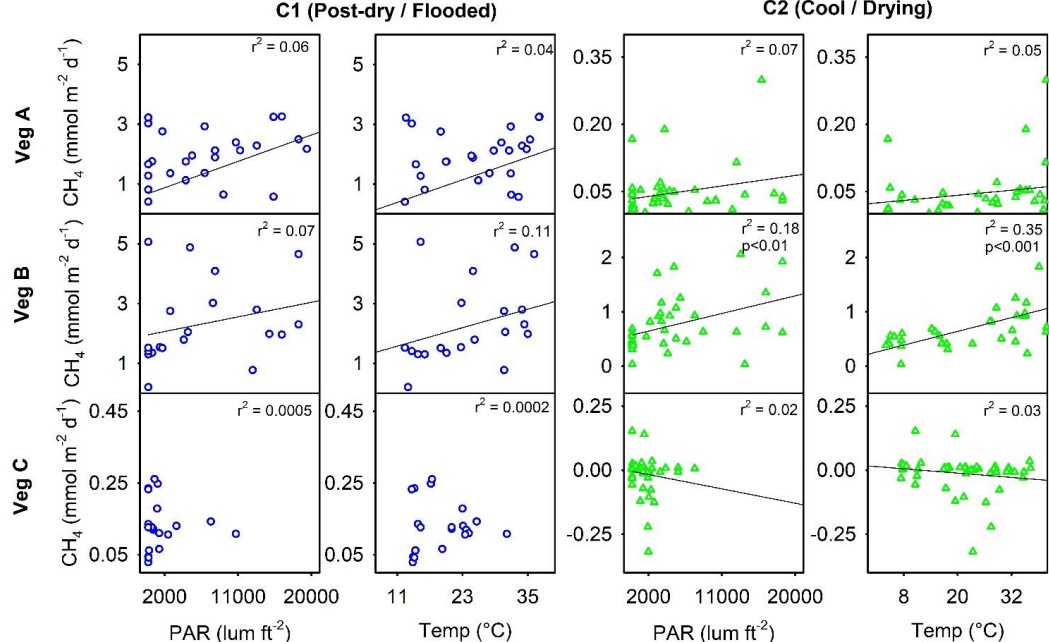


**Figure 6.** Correlations of $CH_4$ with temperature (°C) and photo-synthetically active radiation

(PAR) (lum ft$^{-2}$) for the three seasonal wetland vegetation sites of Cattai Wetland during two

seasonal campaigns.


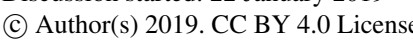


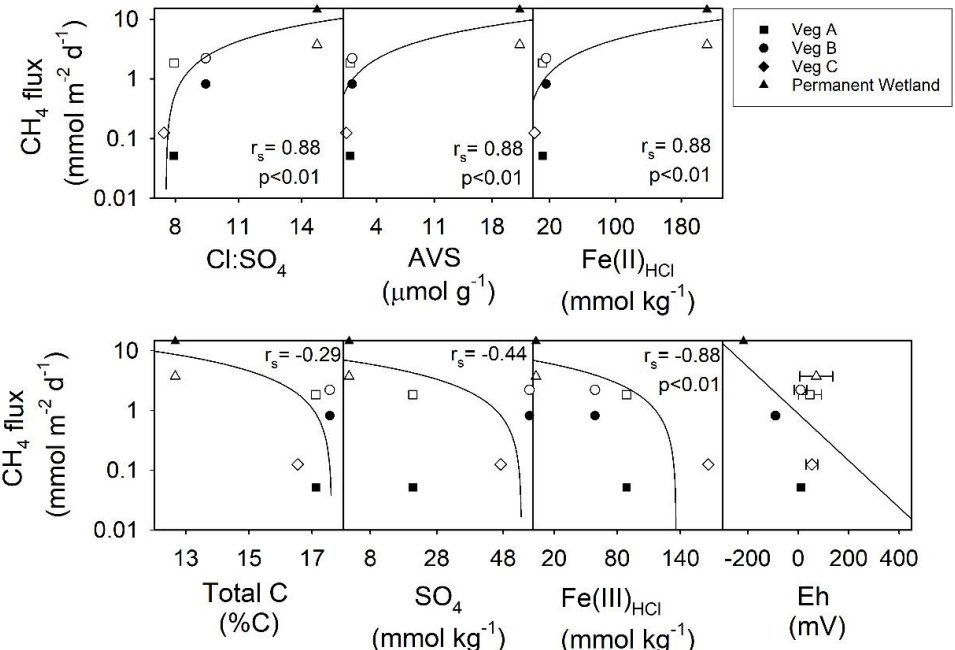


**Figure 7.** Regression analysis of average daily $CH_4$ fluxes (mmol m$^{-2}$ d$^{-1}$) vs subsoil parameters of 0-20 cm core depth (i.e. $CH_4$ 'active' zone). Note: Log scale y-axis of $CH_4$ fluxes from the four wetland ecotypes over two seasons. Note: The $r_s$ values calculated using Spearman rho are for C1 (black shapes) and C2 (white shapes).




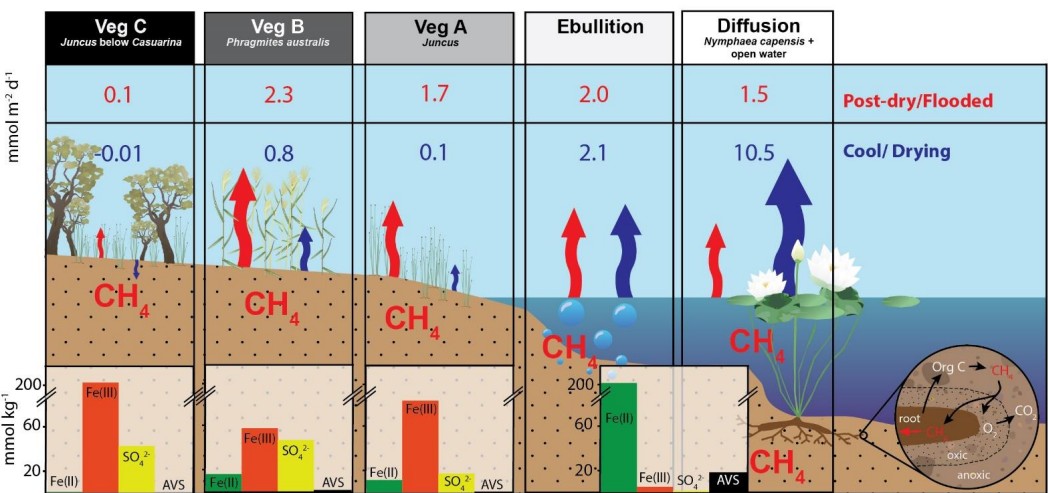

**Figure 8.** Conceptual model summarising the terrestrial and aquatic CH₄ fluxes (mmol m⁻² d⁻¹) and sediment core profile parameters (mmol kg⁻¹) of the permanent and seasonal wetlands during C1 (post-dry/flooded conditions) and C2 (cool/drying conditions) of Cattai Wetland. Conceptual diagram expanded from Jeffrey et al. (in publication) and rhizome process insert adapted from (Conrad, 1993).







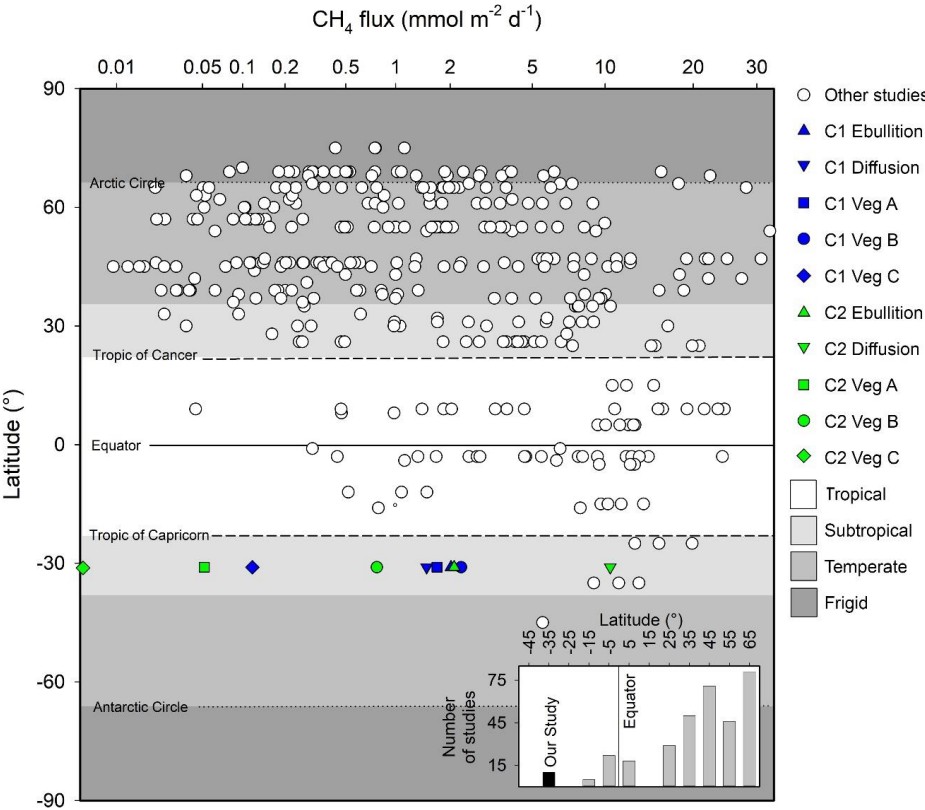

**Figure 9.** Summary of major $CH_4$ wetland reviews by Bartlett and Harriss (1993), Bastviken
et al. (2011) and modelled fluxes by Cao et al. (1998) adapted from Jeffrey et al., (in
publication) highlighting latitudinal trends and bias from a variety of wetland systems. Inset
figure highlights number of studies in these reviews by latitudinal increments of 10° poleward
of the equator. Note: x axis scaled to highlight subtle differences between studies.






**Table 1.** Summary of plant-mediated $CH_4$ fluxes from the seasonal wetland time series and
diurnal $CH_4$ fluxes and ebullition from the permanent wetland during C1 (post-dry/ flooded)
and C2 (cool/ drying). The corresponding sediment core data are average concentrations from
0 to 20 cm below ground level.

| | Permanent Wetland | | Seasonal Wetland Sites | | |
| --- | --- | --- | --- | --- | --- |
| $CH_4$ flux (mmol m$^{-2}$ d$^{-1}$) | Ebullition | Diffusion | Veg A | Veg B | Veg C |
| C1 - Sediment flux | | | 0.06 | 0.04 | 0.10 |
| C1 - Day time | | 0.57 | 1.79 | 2.64 | 0.13 |
| C1 - Night time | | 2.07 | 1.50 | 1.59 | 0.10 |
| C1 - Daily average | 2.02 | 1.49 | 1.70 | 2.27 | 0.12 |
| C2 - Sediment flux | | | 0.0004 | 0.20 | 0.0003 |
| C2 - Day time | | 11.72 | 0.06 | 0.94 | 0.13 |
| C2 - Night time | | 8.39 | 0.04 | 0.48 | 0.10 |
| C2 - Daily average | 2.10 | 10.46 | 0.05 | 0.77 | -0.01 |
| Sediment core average (0-20cm) | | | | | |
| $Fe_{HCl}$ (II) (mmol kg$^{-1}$) | | 202.3 | 11.6 | 15.4 | 1.5 |
| $Fe_{HCl}$ (III) (mmol kg$^{-1}$) | | 5.6 | 83.3 | 56.1 | 204.0 |
| $SO_4^{2-}$ (mmol kg$^{-1}$) | | 1.5 | 17.6 | 45.4 | 43.3 |
| $Cl{:}SO_4^{2-}$ | | 14.8 | 8.4 | 13.9 | 7.4 |
| AVS (µmol g$^{-1}$) | | 18.5 | 0.7 | 0.9 | 0.3 |
| TOC (% C) | | 11.6 | 14.3 | 14.8 | 14.6 |
| C1 - Redox Eh (mV) | | 71.7 | 46.5 | 9.6 | 54.4 |
| C2 - Redox Eh (mV) | | -216 | 12 | -89 | 424 |






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
