# Peer review of "Rhizosphere to the atmosphere: contrasting methane pathways, fluxes and geochemical drivers across the terrestrial-aquatic wetland boundary"

_Biogeosciences, 2019_

## Referee Comment (RC1) · Anonymous Referee #1 · 14 Feb 2019

Review of manuscript bg-2019-11

This study address multiple types of CH4 emissions in wetlands (ebullition, diffusion and plant-mediated flux), their temporal variability (diurnal cycles and seasonal differences), the spatial variability among four wetland vegetation communities in both permanent och seasonal wetlands, and links to wetland soil properties. Hence, it stands out as a potentially valuable study for improved understanding of wetland CH4 emissions. However, I have some concerns and questions below that I think should be addressed

[Figure]

General comments:

It would be good to early on clarify that the word wetland is here used in a broad sense including both wet vegetated environments and open waters/lakes.

L 160 and elsewhere: In warm environments, bubbling can sometimes happen rather continuously leading to very high R2 values (I have experience this myself several times in the tropics). Given the short measurement periods and the very high flux rates sometimes found from the floating chambers, I wonder if they did not received considerable bubbling in such a continuous way leading to linear increase in the headspace. The high variability in the diffusive flux in Fig 3 also seem to support this guess. Are there any data on surface water concentrations of CH4 that could be used together with modelled piston velocities to estimate diffusive flux, or are there any other independent data to verify the high fluxes found as diffusion fluxes? If not, I would hesitate to report the very high fluxes (up to 10 mmol m-2 d-1) as diffusion and I would instead report values from flux chambers as total open water flux including both diffusion and ebullition. This would be a minor loss for the manuscript, compared to the risk of considerably overestimating diffusive fluxes.

I think that it is difficult to claim that this study cover seasonal differences for the CH4 emissions, which are known to have a high day-to-day variability, because there seems to have been on measurement day per season only.

Specific comments:

Abstract: Please define "AVS".

L84-86: Tiny language thing: Two "now" in same sentence.

P156-158. How many replicate floating chamber measurements were performed during each measurement time at each location, and how many measurements times during each campaign?

L185: 10 minute intervals in the daytime sampling would return in the order of 4-6 measurements per hour, but Figure 4 does not show that many points. Were fluxes really measured ad 10 min intervals as said here? L226-230: Please show unit and value of R, as there are several versions to choose from. Should there not be a conversion from ppm to partial pressure in the equation, e.g. s*(1/1000000)*Total_Pressure?

Given the variability, was there really a significant difference between day and night?

L264-265: This statement does not seem to hold for Veg C right?

L265-266: See above: Was there a significant diel variability?

Line 267-268: Is the Veg C flux really negative or rather not significantly different from zero, ie Veg C flux is to be seen as zero?

L269-271: See above comment. I think data and its variability indicate the the floating chambers received lots of ebullition in spite of the gas accumulation being linear. Please provide independent evidence supporting that numbers represent diffusive flux only, or consider reporting fluxes as total flux.

L275: I do not follow the end of this sentence and do not see how Figure 4 can support this statement.

L330 and elsewhere: Is re-flooding the only possible explanation of the differences found in the redox between the seasonal and the permanent wetland? Could not the difference also represent a difference between areas with emergent aquatic plants having O2 leaking out from the roots and maintaining oxidized conditions, and on the other hand areas without this type of root zone aeration in the permanent wetland? This root zone aeration is mentioned below in another context. Should it not also be highlighted here when discussion the sediment redox depth profiles?

L 387-389 and elsewhere: Some studies have highlighted different patterns. See e.g. Milberg et al. 2017 AoB Plants. doi.org/10.1093/aobpla/plx029

L410-411 and elsewhere: Is the difference between passive and pressurised gas transfer the only possibility? The sediment redox potentials reported correlate with CH4 fluxes. Could the sediment conditions not also be influenced also by root depth or root density varying between plant species? If there are no clear explanations, and speculations are necessary, it would be good to highlight not only one alternative (that are frequently discussed in the literature) but also other possible alternatives.

L412-413 and elsewhere: See above. Another perspective could be that that no significant CH4 fluxes were found from the Veg C site. I suggest letting the statistics decide the perspective.

L419-425: Why is not possibly more extensive root zone aeration by the additional tree roots mentioned as one hypothesis?

L428-429: See above. (a) Consider the possibility that the floating chambers reflect total flux and not diffusion only. (b) I am not convinced this study can make claims about seasonal differences based on one measurement day per season only as day-to-day fluxes can be highly variable. Therefore, parts of the discussion about reasons for the seasonal difference seem obsolete.

L451: I suggest removing "Permanent" here, because many large non-permanent wetland areas are also important (most tropical wetlands vary greatly in size over a year).

Fig 1 and elsewhere: Why were not all measurements and core collections taking place nearby eachother? How comparable are the results if data were collected far apart?

Figure 4 and elsewhere: (a) Does Fig 4 really show seasonal fluxes? Can at all seasonal fluxes be claimed from two measurement days as shown here? How to know that these two days were representative of whole seasons? (b) Please inform readers how many replicate measurements were made at each time point for the fluxes?

―――――――――――――――――――

---

## Referee Comment (RC2) · Anonymous Referee #2 · 5 Mar 2019

I apologize for the delay in getting my review in. Overall I think this study is interesting and the paper is mostly well written. It is unfortunate that it is just 2 seasonal sampling events. It is unclear if the sampling occurred over more than one day each season? Please clarify. The authors do a good job of limiting their results to what they can say with the data at hand (assuming that they sampled more than one day per season). I do think that some things need to be clarified. Below I provide comments and suggestions of issues that need to be clarified.

Lines 35 and 36- negative relationships between Fe, and SO4, and CH4? Or Fe and

[Figure]

CH4, and SO4 and CH4? The wording is unclear.

Line 46- what do you mean by early system recovery periods? Recovery from what? Was this wetland recovering from something? This was a remediated wetland?

Line 59- how are drivers and effects of seasonal weather oscillations different?

Line 62- See problems with Mitsch et al 2013 calculations from Bridgham et al 2014 and Neubauer 2014 papers. I see you cite those papers.

Line 68- "lack of spatially resolved wetland CH4 emission data"? There are many studies that have measured this. Some of which you already cited.

Line 84- Is Lal 2008 an appropriate citation for this sentence?

Line 92-how was that 1.2 Pg C estimated?

Line 112- why do you expect the fluxes going to differ across the wetland communities?

Line 162-163- Why were those fluxes reported elsewhere? Is that paper available?

Line 164- how many chambers did you have in each vegetation type? How many days did you measure fluxes? Was it only one day each season?

Lines 279-281- This sentence is more Discussion.

Line 302- Structuring the Discussion in the same order as the Results makes it easier for the reader. I suggest you Discuss your results in the same order they were presented in the Results section.

Line 326-CASS wetland restoration the same as remediation?

Lines 345 and 346- It gets hard to keep track of C1, C2, Veg A, Veg B. Could there be more straightforward ways of talking about these?

Line 352-354- This is more of a results sentence and I am not sure I understand what you are saying. Please clarify.

Line 409 and 410- this Veg A and B is getting tiring. Why not just talk about the species?

Line 433- it is hard for readers to access submitted papers. Please do not cite papers that are not already published in some form.

Line 467- Again see Bridgham et al. 2014 about the problems with Mitsch et al use of radiative forcing vs balance.

Line 452, comma between budget and however.

Figure 6- those are really low r2 values! Are these significant relationships? If they are not significant, it is better not to report the value. And r2 of 0.0005 is better to just say there was no relationship.

Figure 8- I really like this figure. Is Fe(III) in Veg C above the axis break? It is a little hard to tell.

---

## Author Comment (AC1)

Review of manuscript bg-2019-11

This study address multiple types of CH4 emissions in wetlands (ebullition, diffusion and plant-mediated flux), their temporal variability (diurnal cycles and seasonal differences), the spatial variability among four wetland vegetation communities in both permanent och seasonal wetlands, and links to wetland soil properties. Hence, it standsout as a potentially valuable study for improved understanding of wetland CH4 emissions. However, I have some concerns and questions below that I think should be addressed

We thank reviewer 1 for their constructive comments and suggestions, we have responded to each of these comments in blue font below.

General comments:

It would be good to early on clarify that the word wetland is here used in a broad sense including both wet vegetated environments and open waters/lakes.

We agree, this now reads (lines 52-54):

"Wetlands are considered one of the most valuable ecosystems on Earth (Costanza et al., 2014) and may be classified as both permanently inundated (i.e lakes and shallow waters) and seasonally inundated (i.e. vegetated) biomes."

L 160 and elsewhere: In warm environments, bubbling can sometimes happen rather continuously leading to very high R2 values (I have experience this myself several times in the tropics). Given the short measurement periods and the very high flux rates sometimes found from the floating chambers, I wonder if they did not received considerable bubbling in such a continuous way leading to linear increase in the headspace.

We agree this can occur. However, we are also confident that we were able to detect discrete ebullition. For example, our companion paper now published (*Jeffrey, L. C., Maher, D. T., Johnston, S. G., Kelaher, B. P., Steven, A. and Tait, D. R. (2019), Wetland methane emissions dominated by plant-mediated fluxes: Contrasting emissions pathways and seasons within a shallow freshwater subtropical wetland. Limnol Oceanogr. doi:10.1002/lno.11158*) focuses solely on aquatic emissions and provides examples (Fig. S3 – see below) of disregarded floating chambers featuring ebullition bubbles.

[Figure]

**Figure S3.** Examples of $CH_4$ flux linear regression $r^2$ vs flux rate (ppm/sec) for each campaign (left column) and samples of floating chamber $CH_4$ fluxes showing increasing $CH_4$ concentration (ppm) vs time (per second) indicating a successful incubation (top right panel) and an excluded incubation (bottom right panel) due to ebullition bubble release.

To identify this in the manuscript we have added the following:

*"One chamber measurement was removed as an outlier (as it was more than three times the standard deviation of the mean) and any chambers capturing ebullition bubbles (determined by a nonlinear increase in concentration) were also disregarded, see example in Jeffrey et al. (2019)."*

The high variability in the diffusive flux in Fig 3 also seem to support this guess. Are there any data on surface water concentrations of CH4 that could be used together with modelled piston velocities to estimate diffusive flux, or are there any other independent data to verify the high fluxes found as diffusion fluxes? If not, I would hesitate to report the very high fluxes (up to 10 mmol m-2 d-1) as diffusion and I would instead report values from flux chambers as total open water flux including both diffusion and ebullition. This would be a minor loss for the manuscript, compared to the risk of considerably overestimating diffusive fluxes.

We have recently published a companion paper focused on diffusion, ebullition and plant mediated fluxes from the same site using the same techniques (Jeffrey et al. 2019). In this companion paper we assessed water column concentrations, chamber-derived diffusive fluxes, and calculated convection-driven fluxes. That study highlighted that there was both temporal variability in water column CH4 concentrations (CH$_4$ ranging from ~60 uM to 250 uM over a diurnal cycle), and also spatial variability with water column CH$_4$ ranging from 7 to 254 uM throughout the wetland. We also found that convection (occurring only during night) could enhance the piston velocity by up to 17%.

It is likely that this spatial and temporal variability in water column concentrations is the main driver of the observed variability on our current chamber flux estimates. We do not have water column CH4 concentrations from the field campaigns in this present study - however, considering the extremely high water column CH$_4$ concentrations observed in our companion paper (averaging ~ 80 uM), a diffusive flux rate estimate of 10 mmol/m2/d is not extreme and would only require a piston velocity of ~ 0.5 cm/hr. This is piston velocity is similar to the diffusive transfer velocities in wetland measured by deliberate gas tracer experiments (e.g. Ho et al., 2018).

I think that it is difficult to claim that this study cover seasonal differences for the CH$_4$ emissions, which are known to have a high day-to-day variability, because there seems to have been on measurement day per season only.

We agree. We accounted for high resolution measurements however these were snapshots in time. We have removed reference to our fluxes representing 'seasonal' differences from the following lines:

**Abstract (lines 32-34):** "*We account for aquatic CH$_4$ diffusion and ebullition rates, and plant-mediated CH$_4$ fluxes from three distinct vegetation communities, thereby examining diurnal and intra-habitat variability*"

**Lines 346-351:** "*Figure 5. Fluxes of CH$_4$ from diel sampling and ebullition over two campaigns from the permanent wetland and adjacent 24 h time series of the seasonal wetland vegetation types. Note: Diffusive fluxes during C2 include chambers featuring lilies, dashed line represents the average, solid line represents the median and dots represent 5$^{th}$ and 95$^{th}$ percentiles. Letters show groups that did not differ significantly (p>0.05) using ANOVA on ranks and Dunn's pairwise comparisons within each campaign.*"

**Lines 369-370:** "*Figure 6. Correlations of CH$_4$ with temperature (ºC) and photo-synthetically active radiation (PAR) (lum ft$^{-2}$) for the three wetland vegetation sites of Cattai Wetland during two field campaigns.*"

**Lines 408-409:** "*This was associated with the lowest fluxes of CH$_4$ for both sampling periods (Fig. 5, Table 1).*"

**Lines 467-469:** "*These were similar to our findings with highest CH$_4$ fluxes of each campaign time series occurring near midday (10:50 am during C1; 4.88 mmol m$^{-2}$ d$^{-2}$ and 12:15 pm during C2; 2.06 mmol m$^{-2}$ d$^{-2}$) (Fig. 3).*"

**Lines 554-556:** "*Our CH$_4$ emissions rates were at the low end of the scale of measurements made in southern hemisphere subtropical systems but within range of northern hemisphere subtropical systems of similar latitudes (Fig. 9).*"

**Lines 593-597: Conclusion:** "*Results reveal distinct differences between the areal CH$_4$ fluxes of four different eco-types located within a remediated subtropical Australian wetland and indicate high*

*variability between campaigns. By combining novel and well established techniques we delineated several CH$_4$ pathways of both seasonal and permanent wetland sources (ebullition, diffusion and plant-mediated pathways) and linked these to hydrological drivers."*

Specific comments:

Abstract: Please define "AVS".

Amended.

L84-86: Tiny language thing: Two "now" in same sentence.

Amended.

P156-158. How many replicate floating chamber measurements were performed during each measurement time at each location, and how many measurements times during each campaign?

We have added details as follows (Lines 175-177):

 *"A total of 39 CH$_4$ floating chamber incubations averaging ~8 minutes in duration were recorded over the two campaigns, with 19 during C1 (nine at night) and 30 during C2 (12 at night)."*

L185: 10 minute intervals in the daytime sampling would return in the order of 4-6 measurements per hour, but Figure 4 does not show that many points. Were fluxes really measured at 10 min intervals as said here?

We agree this was potentially confusing, this was the approximate intervals between incubation start times. The manuscript stated incubation times '*Vegetation incubation times ranged from 6 to 15 minutes*'. To clarify the number of vegetation incubations measured each day and night, per campaign, we have re-worded this paragraph as follows (lines 205-212):

"*During the first time-series (C1), an average of 16.7 ± 2.9 daytime flux measurements (i.e. after sunrise) and 7.3 ± 1.6 night time (i.e. after sunset) were recorded within each habitat. During the second campaign (C2) an average of 27.7 ± 2.9 (day time) and 10.3 ± 1.5 (night time) flux measurements were recorded within each habitat. In addition, CH$_4$ fluxes from the adjacent exposed sediments or shallow overlying water at each site were also measured at ~4 hourly intervals to determine the influence and role of plant-mediated CH$_4$ fluxes compared to non-vegetated CH$_4$ fluxes....*"

L226-230: Please show unit and value of R, as there are several versions to choose from. Should there not be a conversion from ppm to partial pressure in the equation, e.g. s*(1/1000000)*Total_Pressure?

R is in the units of m$^3$.atm.K$^{-1}$.mol$^{-1}$, which has a value of 8.205*10$^{-5}$ in this case. Text has been added to clarify this point. We assume atmospheric pressure is 1 atm in our calculations, this has been added to the methods section (Lines 251-256):

"        $F = (s(V/RT_{air}A))t$                                           (1)

*where s is the regression slope for each chamber incubation deployments (ppm sec$^{-1}$), V is the chamber volume (m$^3$), R is the universal gas constant (8.205 x 10$^{-5}$ m$^3$.atm.K$^{-1}$.mol$^{-1}$), T$_{air}$ is the air temperature inside the chamber (K), A is the surface area of the chamber (m$^2$) and t is the conversion factor from seconds to day, and to mmol. We assume that atmospheric pressure is 1 atm.*"

Given the variability, was there really a significant difference between day and night?

We have performed statistical analysis to assess differences between day and night fluxes and have added the following to the manuscript methods (Lines 262-266):

*"2.6 Statistical analysis*

*As the $CH_4$ flux data was non-parametric we used a Kruskal-Wallis one way analysis of variance (ANOVA) on ranks to test for significant differences between each campaign, between flux pathways and between diel variability, where $p<0.001$. Dunn's multiple pairwise comparisons were then used to analyse specific sample pairs ($p<0.05$)."*

And abstract (lines 39-39):

…*"Significantly higher $CH_4$ emissions ($p<0.001$) of the seasonal wetland were measured during flooded conditions…"*

And to our results (lines 334-351):

*"$CH_4$ fluxes from the three vegetation types were significantly higher during C1 than during C2 ($p<0.001$). During C1, the $CH_4$ fluxes from the Juncus and Phragmites were not significantly different from each other but were both significantly higher ($p<0.001$) than Juncus/Forest however, during C2 the $CH_4$ fluxes of each seasonal wetland habitat were significantly different between all habitats ($p<0.05$) (Fig. 5). The highest average $CH_4$ fluxes in each of the vegetation types always occurred during the daytime but were not significantly different to night time fluxes (Fig. 5, Table 1). Phragmites consistently emitted the highest $CH_4$ fluxes (2.27 ± 1.42 mmol $m^{-2}$ $d^{-1}$ during C1 and 0.77 ± 0.46 mmol $m^{-2}$ $d^{-1}$ during C2). The Juncus/ Forest ecotype within the seasonal wetland consistently produced the lowest $CH_4$ fluxes of all sites, with a negligible flux that was not significantly different from zero occurring during C2 (-0.01 ± 0.08 mmol $m^{-2}$ $d^{-1}$)."*

[Figure]

**Figure 5.** *Fluxes of CH$_4$ from diel sampling and ebullition over two campaigns from the permanent wetland and adjacent 24 h time series of the seasonal wetland vegetation types. Note: Diffusive fluxes during C2 include chambers featuring lilies, dashed line represents the average, solid line represents the median and dots represent 5$^{th}$ and 95$^{th}$ percentiles. Letters show groups that did not differ significantly (p>0.05) using ANOVA on ranks and Dunn's pairwise comparisons within each campaign.*

L264-265: This statement does not seem to hold for Veg C right?

After statistical analysis, this now reads (Lines 338-340):

*"The highest average CH$_4$ fluxes in each of the vegetation types always occurred during the daytime but were not significantly different to night time fluxes (Fig. 5, Table 1)."*

L265-266: See above: Was there a significant diel variability?

As address above this now reads (lines 338-341):

*"The highest average CH$_4$ fluxes in each of the vegetation types always occurred during the daytime but were not significantly different to night time fluxes (Fig. 5, Table 1). Phragmites consistently emitted the highest CH$_4$ fluxes (2.27 ± 1.42 mmol m$^{-2}$ d$^{-1}$ during C1 and 0.77 ± 0.46 mmol m$^{-2}$ d$^{-1}$ during C2)...."*

Line 267-268: Is the Veg C flux really negative or rather not significantly different from zero, ie Veg C flux is to be seen as zero?

As the flux is nominal we have re-worded as (lines 341-344):

*"…The Juncus/ Forest ecotype within the seasonal wetland consistently produced the lowest $CH_4$ fluxes of all sites, with a negligible flux that was not significantly different from zero occurring during C2 (-0.01 ± 0.08 mmol $m^{-2}$ $d^{-1}$)."*

L269-271: See above comment. I think data and its variability indicate the floating chambers received lots of ebullition in spite of the gas accumulation being linear. Please provide independent evidence supporting that numbers represent diffusive flux only, or consider reporting fluxes as total flux.

In addition to our earlier response addressing this point and our evidence supporting that the reported data represent diffusive flux only (see response to General comment no. 2), we note that the rates are within those reported in previously published studies of diffusive fluxes from similar latitudes, and thus are representative of both open water and lilies.

The higher fluxes during C2 are also likely due to the re-emergence of lily species (Nymphaea sp.) during C2 which are included in some chamber measurements (but were not present not C1). These were mentioned in the manuscript, but we have now added further details to the following areas to clarify for the reader (lines 346-351):

*"**Figure 5.** Fluxes of $CH_4$ from diel sampling and ebullition over two campaigns from the permanent wetland and adjacent 24 h time series of the seasonal wetland vegetation types. Note: Diffusive fluxes during C2 include chambers featuring lilies, dashed line represents the average, solid line represents the median and dots represent $5^{th}$ and $95^{th}$ percentiles. Letters show groups that did not differ significantly (p>0.05) using ANOVA on ranks and Dunn's pairwise comparisons within each campaign."*

*lines 353-357:*

*…" The permanent wetland showed an inverse trend with seven-fold and significantly higher (p<0.001) diffusive fluxes during the cool/drying C2 when lilies were present (10.46 ± 15.81 mmol $m^{-2}$ $d^{-1}$) compared to the post-dry/flooded C1 when no lilies were present (1.49 ± 2.75 mmol $m^{-2}$ $d^{-1}$), while the ebullition rates were similar during both campaigns (Fig. 5, Table 1).….."*

*lines 326-335:*

*"…A lag time (ranging from weeks to months) for recovery of the $CH_4$ pool post-drought has been observed in other systems (Boon et al., 1997) and also during lab-based experiments (Knorr et al., 2008; Freeman et al., 1992). Further, during C2 the return of macrophyte species Nymphaea caspensis most likely enhanced $CH_4$ gas transport from the rhizosphere to the floating chambers, as discussed in detail in Jeffrey et al. (2019). Therefore this combination of drivers most likely explain the higher $CH_4$ fluxes during C2 when the system (and lilies) had sufficient time to recover, despite lower water column temperatures that would normally reduce microbial metabolism rates. This hypothesis is also supported by the shift of net positive redox potential…"*

*Lines 171-175:*

*"To account for spatial and temporal variability, measurements were conducted during both day-time and night-time, and sampling within vegetated areas featuring lilies (Nymphaea capensis); that*

*were only present during the second campaign, forested areas (Melaleuca sp.) and in areas where no aquatic vegetation was present (i.e. open water)."*

L275: I do not follow the end of this sentence and do not see how Figure 4 can support this statement.

We agree. We have amended this sentence, incorporated new results from the ANOVA as follows (lines 357-360):

*"Overall, the diffusive fluxes of the permanent wetland were within range of $CH_4$ fluxes from the three seasonal wetland habitats but were significantly higher than Juncus/Forest during both campaigns, and Juncus during C2 (Fig. 5). Diel diffusive flux variability was not significant between day time and night time (Table. 1, Fig. 5)."*

L330 and elsewhere: Is re-flooding the only possible explanation of the differences found in the redox between the seasonal and the permanent wetland? Could not the difference also represent a difference between areas with emergent aquatic plants having O2 leaking out from the roots and maintaining oxidized conditions, and on the other hand areas without this type of root zone aeration in the permanent wetland? This root zone aeration is mentioned below in another context. Should it not also be highlighted here when discussion the sediment redox depth profiles?

Although we agree this is another plausible explanation, especially for the seasonal wetland sites, it is unlikely to apply for the permanent wetland, as the *opposite trend* occurred due to the absence of lilies during C1; where the positive redox potentials were observed. During C2 when the lilies returned, lower redox was observed. To clarify this point, in the permanent wetland discussion we have added: (lines 537-540)

*"…Further, although aquatic vegetation can facilitate root zone aeration therefore increasing sedimentary redox potential, as no aquatic vegetation was present in the permanent wetland during C1, this further suggests water level drawdown of the was the main driver of redox conditions."*

And to the seasonal wetland discussion we have added the following text (lines 510-514):

*"The differences are therefore likely explained by the higher positive redox potentials (Table 1) that may be partially attributable to rhizome aeration by the nearby trees, and more abundant thermodynamically favourable terminal electron acceptors (i.e. Fe(III) and $SO_4^{2-}$) (Fig. 5) all of which can inhibit methane production within the sediments (Burdige, 2012)."*

L 387-389 and elsewhere: Some studies have highlighted different patterns. See e.g. Milberg et al. 2017 AoB Plants. doi.org/10.1093/aobpla/plx029

We have added this paper to the discussion as follows (lines 463-467):

*"…Milberg et al. (2017) found no apparent diel patterns of $CH_4$ fluxes from Phragmites australis during seven campaigns within the Swedish growing season. Kim et al. (1998) showed that $CH_4$ emissions peaked around midday and that daytime emissions were about 3-fold higher than night time emissions, positively correlating with temperature and PAR…"*

L410-411 and elsewhere: Is the difference between passive and pressurised gas transfer the only possibility? The sediment redox potentials reported correlate with $CH_4$ fluxes. Could the sediment conditions not also be influenced also by root depth or root density varying between plant species? If there are no clear explanations, and speculations are necessary, it would be good to highlight not

only one alternative (that are frequently discussed in the literature) but also other possible alternatives.

We agree on the need to canvas a wider range of possible explanations and have now discussed potential alternatives as follows (lines 490-499):

*"In comparison, in Phragmites these day:night ratios were almost triple this (67% and 94% higher) during the same periods. This may potentially be due to the more efficient daytime conductive gas transfer pathway of $CH_4$ through Phragmites australis compared to the more passive diffusive $CH_4$ gas transfer pathway of Juncus kraussii and/or the effectiveness of these different species to alter sedimentary redox conditions. This suggests that non-pressurized pathways may result in lower net rhizosphere-atmosphere gas exchange of $CH_4$ from seasonal wetland vegetation. Alternatively, root depth and root density differ between these two species (Moore et al., 2012, De La Cruz et al., 1977), therefore further influencing redox dynamics in the rhizosphere, and the potential extent of net gas exchange."*

L412-413 and elsewhere: See above. Another perspective could be that that no significant CH4 fluxes were found from the Veg C site. I suggest letting the statistics decide the perspective.

As now addressed in the results, we have added 'significant' to this as follows (lines 500-501):

*"The Juncus/ Forest habitat emitted significantly lower fluxes of $CH_4$ during both time series campaigns and was a net sink for $CH_4$ during C2 (Table 1, Fig. 8)..."*

L419-425: Why is not possibly more extensive root zone aeration by the additional tree roots mentioned as one hypothesis?

As mentioned above, we have added to this hypothesis as follows (lines 507-514):

*"Shading by the overhanging trees may inhibit the daytime diffusive $CH_4$ gas transport through Juncus/ Forest habitat assumable to lower rates of photosynthesis, however PAR was only lower during C2 (Fig. 7) and so does not appear to explain the $CH_4$ flux differences observed during C1. The differences are therefore likely explained by the higher positive redox potentials (Table 1) that may be partially attributable to rhizome aeration by the nearby trees, and more abundant thermodynamically favourable terminal electron acceptors (i.e. Fe(III) and $SO_4^{2-}$) (Fig. 5) all of which can inhibit methane production within the sediments (Burdige, 2012)."*

L428-429: See above. (a) Consider the possibility that the floating chambers reflect total flux and not diffusion only. (b) I am not convinced this study can make claims about seasonal differences based on one measurement day per season only as day-to-day fluxes can be highly variable. Therefore, parts of the discussion about reasons for the seasonal difference seem obsolete.

(a) As per previous comments, we are confident that reported diffusion values are accurate and likely due to the presence of lilies enhancing the flux as mentioned in detail above. (b) As per previous suggestions and reviewer #2 comments also, we have removed all claims to quantifying 'seasonal fluxes' from the manuscript and stick to the changes in drivers in our discussion.

L451: I suggest removing "Permanent" here, because many large non-permanent wetland areas are also important (most tropical wetlands vary greatly in size over a year).

Removed and this now reads (lines 546-548):

"Within the global wetland $CH_4$ budget both subtropical systems and southern hemisphere systems are poorly represented (Bartlett and Harriss, 1993;Bastviken et al., 2011) (Fig. 9)."

Fig 1 and elsewhere: Why were not all measurements and core collections taking place nearby each other? How comparable are the results if data were collected far apart?

At the seasonal wetland sites (Veg A, B and C) cores were taken nearby, but not directly at the site of the flux measurements to ensure minimal disturbance of the site.  As the permanent wetland was fairly homogenous (as found during previous study of the wetland i.e. Jeffrey et al., 2019), the cores were extracted from a location to avoid trampling disturbance to fragile sediments and lily habitat, and to avoid artificial ebullition release prior to deployment.  We have added the following to our methods explaining this (lines 222-224):

"The cores were sampled in close proximity to the time series habitats (5 to 15 m) in December 2016, but within the permanent wetland the cores were taken from elsewhere to avoid disturbance of the shallow water column and sediments."

Figure 4 and elsewhere: (a) Does Fig 4 really show seasonal fluxes? Can at all seasonal fluxes be claimed from two measurement days as shown here? How to know that these two days were representative of whole seasons? (b) Please inform readers how many replicate measurements were made at each time point for the fluxes?

We have removed all terms referring our study to 'seasonal fluxes' and replaced with 'campaigns' and as above have referenced our companion study and included the number of chamber measurements featured in this study in our methods (lines 171-178):

"To account for spatial and temporal variability, measurements were conducted during both day-time and night-time, and sampling within vegetated areas featuring lilies (Nymphaea capensis); that were only present during the second campaign, forested areas (Melaleuca sp.) and in areas where no aquatic vegetation  was present (i.e. open water). A total of 39 $CH_4$ floating chamber incubations averaging ~8 minutes in duration were recorded over the two campaigns, with 19 during C1 (nine at night) and 30 during C2 (12 at night). The average $r^2$ value of linear regressions of $CH_4$ concentrations versus time during chamber incubations was 0.97 ± 0.05."

End of Referee #1 response file

---

## Author Comment (AC2)

I apologize for the delay in getting my review in. Overall I think this study is interesting and the paper is mostly well written. It is unfortunate that it is just 2 seasonal sampling events. It is unclear if the sampling occurred over more than one day each season? Please clarify. The authors do a good job of limiting their results to what they can say with the data at hand (assuming that they sampled more than one day per season). I do think that some things need to be clarified. Below I provide comments and suggestions of issues that need to be clarified.

We thank referee #2 for their constructive feedback and suggestions. We have attended to all of these in details below in blue font.

Lines 35 and 36- negative relationships between Fe, and SO4, and CH4? Or Fe and CH4, and SO4 and CH4? The wording is unclear.

Amended as follows (Lines 35-38):

*"For example, distinct negative relationships between $CH_4$ fluxes and both $Fe(III)$ and $SO_4^{2-}$ were observed. Where sediment $Fe(III)$ and $SO_4^{2-}$ were depleted distinct positive trends occurred between $CH_4$ emissions and $Fe(II)$ / acid volatile sulphur (AVS).*

Line 46- what do you mean by early system recovery periods? Recovery from what? Was this wetland recovering from something? This was a remediated wetland?

We meant recovery from anthropogenic drainage. To clarify this for the reader, the abstract now reads as follows (lines 45-47):

*"…We suggest that wetland remediation strategies should consider geochemical profiles to help to mitigate excessive and unwanted methane emissions, especially during early system remediation periods."*

Line 59- how are drivers and effects of seasonal weather oscillations different?

We have added (lines 64-70):

*"Resolving the drivers, pathways and effects of seasonal weather oscillations on wetland $CH_4$ sink or source behaviours is important to enable more accurate climate model projections and to reduce uncertainties in the global wetland $CH_4$ budget (Saunois et al., 2016; Kirschke et al., 2013). Weather oscillations affect the total wetland areal extent and inundation periods, with wet conditions facilitating anaerobic conditions favouring methanogenesis, while the opposite is seen during dry periods which potentially mitigates $CH_4$ emissions (Whiting and Chanton, 2001;Wang et al., 1996)."*

Line 62- See problems with Mitsch et al 2013 calculations from Bridgham et al 2014 and Neubauer 2014 papers. I see you cite those papers.

We agree and have acknowledged Bridgham et al., (2014)'s response paper as follows (lines 70-75):

*"Mitsch et al. (2013) estimated that the average ratio of freshwater wetland $CO_2$ sequestration to $CH_4$ emissions was 25.5:1, though was later refuted by Bridgham et al. (2014). As $CH_4$ is 34 times more potent than carbon dioxide ($CO_2$) over a 100 year time scale (Stocker et al., 2013), this suggests that many freshwater wetlands may have a net positive radiative forcing effect on climate (Petrescu et al., 2015;Hernes et al., 2018)."*

Line 68- "lack of spatially resolved wetland CH4 emission data"? There are many studies that have measured this. Some of which you already cited.

*Agree, by 'spatial' we meant latitudinal as discussed with figure 9. This now reads (lines 77-82):*

*"The lack of latitudinally-resolved wetland $CH_4$ emission data, as well as the limited number of studies constraining the multiple wetland $CH_4$ flux pathways (i.e. ebullition, diffusion and plant-mediated) coupled with ongoing anthropogenic conversion of wetland systems (Saunois et al., 2016;Neubauer and Megonigal, 2015;Bartlett and Harriss, 1993) further contribute to the uncertainties around $CH_4$ regional to global scale budgets."*

 Line 84- Is Lal 2008 an appropriate citation for this sentence?

*Removed.*

Line 92-how was that 1.2 Pg C estimated?

*We have now provided an explanation as follows (lines 99-104):*

*"Within Australia, it has been estimated that more than 50% of natural wetlands have been lost to land use change, drainage and degradation since European settlement (Finlayson and Rea, 1999;ANCA, 1995). By comparing and reviewing pristine Australian wetland carbon stocks to drained sites, and GHG dynamics, Page and Dalal (2011) estimated that through biomass loss, enhanced soil respiration, $N_2O$ production and a reduction in $CH_4$ emissions, that Australian wetland loss equated to ~1.2 Pg $CO_2$ equivalents emitted to the atmosphere."*

Line 112- why do you expect the fluxes going to differ across the wetland communities?

*We have added (lines 122-128):*

*"…We hypothesize that wetland $CH_4$ emissions will differ significantly between the campaigns and between the four wetland communities due to differences in soil chemistry, hydrology and plant physiology. We account for three atmospheric flux pathways for methane; ebullition, diffusion and plant-mediated fluxes, over diurnal cycles and within different hydrological conditions. $CH_4$ fluxes were also assessed in relation to the underlying soil properties, including sulphate, reactive iron III and iron II, acid volatile sulphur, chloride and organic carbon."*

Line 162-163- Why were those fluxes reported elsewhere? Is that paper available?

*This is a companion study that is now published. This passage is now updated as (lines 181-183):*

*"…Examples of these, in addition to the ebullition and diffusive $CH_4$ flux methods and measurements from the permanent wetland have previously been reported elsewhere (Jeffrey et al., 2019)."*

Line 164- how many chambers did you have in each vegetation type? How many days did you measure fluxes? Was it only one day each season?

*We sampled in triplicate within each vegetation type (three sites), so n=9 in total. We measured at least a complete diel cycle each season. Each campaign covered five days with ebullition deployments, diel diffusion rates and redox etc. To more clearly clarify this in the manuscript we have added the following lines 186-191:*

*"Simultaneous time series chamber experiments were conducted over a minimum of 24 hours to measure diel $CH_4$ fluxes during each season from the three different wetland vegetation ecotypes. These ecotypes were Juncus kraussii, Phragmites australis and Juncus kraussii amongst Casuarina sp. forest (Fig. 1). In each ecotype, three acrylic bases (65 x 65 x 30 cm) were installed four months before the first time series experiment, to minimise disturbance to the sediment profile and vegetative rhizosphere.*

And as per referee #1 suggestions we have also included more details about the number of chamber measurements taken during each campaign as follows at lines 204-212:

*"Vegetation incubation times ranged from 6 to 15 minutes depending on the flux rate and were taken from triplicate chambers to account for heterogeneity within each ecotype. During the first time-series (C1), an average of 16.7 ± 2.9 daytime flux measurements (i.e. after sunrise) and 7.3 ± 1.6 night time (i.e. after sunset) were recorded within each habitat. During the second campaign (C2) an average of 27.7 ± 2.9 (day time) and 10.3 ± 1.5 (night time) flux measurements were recorded within each habitat. In addition, $CH_4$ fluxes from the adjacent exposed soils or shallow overlying water at each site were also measured at ~4 hourly intervals to determine the influence and role of plant-mediated $CH_4$ fluxes compared to non-vegetated $CH_4$ fluxes."*

And diffusive chamber measurements (lines 175-177):

*"…A total of 39 $CH_4$ floating chamber incubations averaging ~8 minutes in duration were recorded over the two campaigns, with 19 during C1 (nine at night) and 30 during C2 (12 at night)."*

Lines 279-281- This sentence is more Discussion.

We agree and have moved to discussion as follows at line 375-379:

*"Sediment profiles provide insights to the historical geochemical changes that have occurred across the CASS landscapes of the four Cattai Wetland sites (Fig. 5). We base our results and discussion on the upper rhizosphere depth zone (20 cm) as this featured the highest organic carbon concentrations is therefore assumed to be an active area of carbon metabolism, and $CH_4$ production and consumption (Nedwell and Watson, 1995)…."*

Line 302- Structuring the Discussion in the same order as the Results makes it easier for the reader. I suggest you Discuss your results in the same order they were presented in the Results section.

We agree and have aligned our results to be the same order as our discussion.

Line 326-CASS wetland restoration the same as remediation?

Now amended to 'remediation' to be consistent.

Lines 345 and 346- It gets hard to keep track of C1, C2, Veg A, Veg B. Could there be more straightforward ways of talking about these?

As per previous suggestions we have renamed our three vegetation sites with more intuitive titles: 'Juncus', 'Phragmites' and 'Juncus/Forest' but will keep C1 and C2 to represent the two campaigns.

Line 352-354- This is more of a results sentence and I am not sure I understand what you are saying. Please clarify.

We agree and have removed as this sentence was redundant and have now combined with the previous paragraph as (lines 422-428):

*"…Further, as iron reduction yields more free energy than $SO_4^{2-}$ reduction (Burdige, 2012), then Fe reduction at the Juncus site may outcompete $CH_4$ production ahead of $SO_4^{2-}$ reduction at Phragmites, which may help explain some of the differences in $CH_4$ production between the two sites. The positive significant trends between Fe(II), AVS and the $Cl:SO_4^{2-}$ ratios with $CH_4$ flux rates ($r_s$=0.88, p<0.01) further support our hypothesis that reducing conditions and a smaller pool of sediment Fe(III) and $SO_4^{2-}$ facilitate higher $CH_4$ production rates (Fig. 7)…."*

Line 409 and 410- this Veg A and B is getting tiring. Why not just talk about the species?

As mentioned above, we have amended throughout the manuscript and adjusted all figures. This are now introduced first by scientific name and then shortened as follows at lines 156-159:

*"The seasonal wetland to the south is dominated by the sedge; Juncus kraussii ('Juncus' from herein) and features scattered stands of Phragmites australis ('Phragmites' from herein) with areas of slightly higher elevation dominated by Juncus kraussii below Casuarina sp. ('Juncus/ Forest' from herein) (Fig. 1)."*

Line 433- it is hard for readers to access submitted papers. Please do not cite papers that are not already published in some form.

These are now published and the reference list has been updated.

Line 467- Again see Bridgham et al. 2014 about the problems with Mitsch et al use of radiative forcing vs balance.

As these numbers have been refuted, we have removed Mitch and provided a more suitable reference to this uncertainty at lines 565-569:

*"Although remediating degraded wetlands through re-flooding is a common technique to improve biodiversity, increase C sequestration and improve downstream water quality issues (Johnston et al., 2014;Johnston et al., 2004), our results propose a nuanced dilemma for land use managers, as wetland remediation can potentially have net positive radiative forcing effects on the Earth's climate due to high rates of $CH_4$ production  (Petrescu et al., 2015)."*

Line 452, comma between budget and however.

Amended.

Figure 6- those are really low r2 values! Are these significant relationships? If they are not significant, it is better not to report the value. And r2 of 0.0005 is better to just say there was no relationship.

Amended.

[Figure]

**Figure 6.** Correlations of CH$_4$ with temperature (ºC) and photo-synthetically active radiation (PAR) (lum ft$^{-2}$) for the three seasonal wetland vegetation sites of Cattai Wetland during two field campaigns.

Figure 8- I really like this figure. Is Fe(III) in Veg C above the axis break? It is a little hard to tell.

We have added a dashed line to show the axis break more clearly as follows:

[Figure]

**Figure 8.** Conceptual model summarising the terrestrial and aquatic CH$_4$ fluxes (mmol m$^{-2}$ d$^{-1}$) and sediment core profile parameters (mmol kg$^{-1}$) of the permanent and seasonal wetlands during C1 (post-dry/flooded conditions) and C2 (cool/drying conditions) of Cattai Wetland. Conceptual diagram expanded from Jeffrey et al. (2019) and rhizome process insert adapted from (Conrad, 1993). Note: Dashed line highlights y-axis break.